# Unlocking Speech Instruction Data Potential with Query Rewriting

## Abstract

End-to-end Large Speech Language Models (**LSLMs**) demonstrate strong potential in response latency and speech comprehension capabilities, showcasing general intelligence across speech understanding tasks. However, the ability to follow speech instructions has not been fully realized due to the lack of datasets and heavily biased training tasks. Leveraging the rich ASR datasets, previous approaches have used Large Language Models (**LLMs**) to continue the linguistic information of speech to construct speech instruction datasets. Yet, due to the gap between LLM-generated results and real human responses, the continuation methods further amplify these shortcomings. Given the high costs of collecting and annotating speech instruction datasets by humans, using speech synthesis to construct large-scale speech instruction datasets has become a balanced and robust alternative. Although modern Text-To-Speech (**TTS**) models have achieved near-human-level synthesis quality, it is challenging to appropriately convert out-of-distribution text instruction to speech due to the limitations of the training data distribution in TTS models. To address this issue, we propose a query rewriting framework with multi-LLM knowledge fusion, employing multiple agents to annotate and validate the synthesized speech, making it possible to construct high-quality speech instruction datasets without relying on human annotation. Experiments show that this method can transform text instructions into distributions more suitable for TTS models for speech synthesis through zero-shot rewriting, increasing data usability from 72% to 93%. It also demonstrates unique advantages in rewriting tasks that require complex knowledge and context-related abilities.

## 1 Introduction

LLMs have demonstrated powerful performance in general intelligence, profoundly changing the way humans interact with AI systems (OpenAI et al., 2024; Abdin et al., 2024; Dubey et al., 2024). However, constrained by the text modality, existing LLMs are unable to meet the needs of rich real-world interactive scenarios, making it a natural idea to extend this general capability to more modalities (Fang et al., 2024). Recently, end-to-end LSLMs have shown great potential in terms of response latency and speech understanding, making it possible to extend this general performance to scenarios better suited for verbal interaction (Chu et al., 2023; 2024; Zhang et al., 2023a; Fathullah et al., 2024). However, due to the lack of high-quality speech instruction datasets and heavily biased training tasks, the ability of LSLMs to follow speech instructions is not fully realized, resulting in a lack of intent perception in verbal interaction scenarios. Thus, building a high-quality large-scale speech instruction dataset becomes a crucial foundation for advancing the ability to follow speech instructions.

Benefiting from abundant Automatic Speech Recognition (**ASR**) datasets (Ardila et al., 2020; Pratap et al., 2020; Guoguo Chen, 2021; Wang et al., 2024), early work aligned speech and text modalities by having models repeat or recognize linguistic information in speech through textual instructions (Gong et al., 2023). To enhance the model's understanding of paralinguistic information in speech, traditional tasks in the speech domain (such as speaker classification, speech entity recognition, etc.) were integrated into training (Chu et al., 2023; Tang et al., 2024; Das et al., 2024). These training methods, which treat the speech modality as files rather than instructions, cause the model to lack the ability to follow speech instructions and lead to hallucinations due to the severe bias in

task distribution. To unlock the model's potential to follow speech instructions, previous approaches constructed speech instruction datasets by continuing the linguistic information of speech through LLMs to restore the model's instruction-following capabilities (Fathullah et al., 2024). However, due to the gap between the generated results of LLMs and human-annotated ground truth answers, using such continuation methods may further amplify these shortcomings.

Unlike the image domain, which has abundant human-annotated data (Laurençon et al., 2024; Lin et al., 2015), constructing large-scale, manually narrated, and annotated speech instruction datasets from scratch is challenging due to the high costs of collection and annotation. As a result, using speech synthesis to construct datasets becomes a robust choice after weighing the pros and cons. However, due to differences between TTS models and human speech narration, as well as hallucination issues during the speech synthesis process, it is necessary to verify whether the synthesized speech is linguistically equivalent to the text. On the other hand, since TTS models have a limited vocabulary, they cannot accurately convert out-of-distribution text, such as compound words, abbreviations, or mathematical formulas, into speech, leading to the loss of linguistic information.

Previous work has shown that the semantic similarity between two texts can be calculated using embedding models, demonstrating better robustness compared to methods like WER and more closely aligned with how humans assess textual similarity (Muennighoff et al., 2023). Some high-quality human-annotated datasets achieve superior annotation results by integrating the opinions of multiple annotators, making the sample distribution more representative of real-world scenarios (Deitke et al., 2024; Liu et al., 2021). Meanwhile, query rewriting optimizes the text distribution without significantly altering the semantics, making it better aligned with a specific distribution (Ye et al., 2024).

Inspired by this, we propose a query rewriting method with multi-LLM knowledge fusion, along with multi-agent annotation and data quality validation. Specifically, we leverage LLMs' generalization ability in zero-shot tasks to guide the rewriting of text instructions to fit the training distribution of TTS models. Additionally, we use multiple distinct LLMs to rewrite the text from different perspectives. Next, we automatically extract linguistic information from the speech using multiple models and calculate its average similarity to the original text in the embedding space to avoid annotation errors and achieve semantically optimal results. Finally, we fuse the knowledge of different LLMs in this zero-shot rewriting task to tackle challenging rewrites that require complex knowledge. Experimental results show that our method increased data usability from 71% to 93% and improved the semantic similarity between the linguistic information in speech and the original text by 5%.

The main contribution of this paper can be summarised as follows:

- We propose a query rewriting framework with multi-LLM knowledge fusion, along with a multi-agent annotation and validation method based on embedding space similarity, enabling the low-cost, automated construction of high-quality speech instruction datasets.

- Experiments show that our proposed method demonstrates unique advantages in rule-based query rewriting, context-aware understanding, and complex knowledge integration, increasing the average data usability from 72% to 93%, while maintaining consistent performance across different validation methods.

- By comparing the training results using voice data of varying quality and alignment objectives, we validated the significant advantages of high-quality synthesized speech data in alignment effectiveness and cross-modal consistency. It also demonstrates the importance of learning from real human responses to enhance the model's ability to follow speech instructions.

## 2 RELATED WORK

**Text-To-Speech** TTS models have recently demonstrated impressive performance in terms of synthesis quality and fluency. However, constrained by the reliance on reference speech, they lack diversity in style (Wang et al., 2023; Le et al., 2023). Inspired by the image modality, using natural language descriptions of speaker styles to address this issue has become a promising approach. However, due to the lack of large-scale speech datasets containing natural language descriptions, it is difficult to freely use natural language to control the style of speech synthesis. Lyth & King (2024)

constructed a large-scale speech dataset using multiple attributes for automated labeling, which significantly improved the synthesis diversity of TTS models. In this work, we utilize GPT-4 (OpenAI et al., 2024) to synthesize style descriptions and use Parler-TTS (Lacombe et al., 2024) to synthesize speech.

**Speech-text alignment training in LSLMs** Aligning speech and text modalities is crucial for building the speech understanding capability of end-to-end LSLMs. Benefiting from abundant ASR datasets (Ardila et al., 2020; Pratap et al., 2020; Guoguo Chen, 2021; Wang et al., 2024), early work aligned the two modalities by instructing the model to repeat or recognize the linguistic information in the speech (Shu et al., 2023; Zhang et al., 2023a; Gong et al., 2023; Chu et al., 2023). However, due to the severely biased task distribution, the model ignored following the instructions in the speech and defaulted to performing language recognition (Fathullah et al., 2024). Recent work has employed a continuation approach to construct speech instruction data, aiming to restore the model's ability to follow spoken instructions (Fathullah et al., 2024; Fang et al., 2024). Due to the gap between the language model's generated outputs and real human responses, using the continuation method to train the model can further amplify this deficiency (Seddik et al., 2024; Chen et al., 2024). In this work, we propose a high-quality speech instruction synthesis method to address this issue.

## 3 PROBLEM FORMULATION

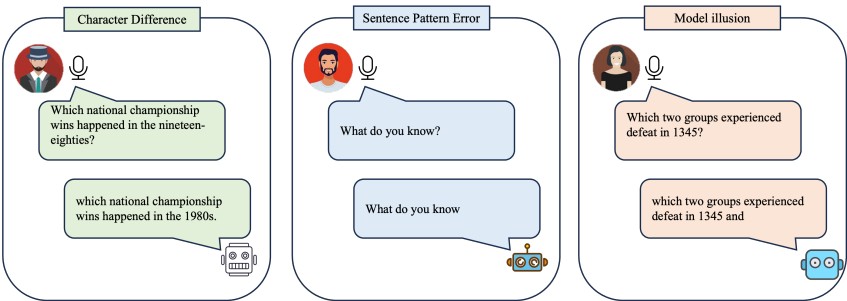

Figure 1: Some recognition errors in ASR models. *Sentence Pattern Error* indicates that the ASR model failed to provide appropriate punctuation based on the user's intent. In the given examples,the original sentence expresses a question, while the ASR-recognized sentence, lacking proper punctuation, conveys a tone of disdain or assertion.

Our goal is to automatically obtain synthesized speech that is linguistically equivalent to the text. For a given TTS model and text instruction $c_o$, we can get the synthesized speech $s_o$.

Ideally, we can get the linguistic information $\bar{c}_o$ in $s_o$ by ASR model, it should satisfies

$$\bar{c}_o = c_o \tag{1}$$

which mean that the $s_o$ is linguistically equivalent to $c_o$ strictly.

As shown in Figure 1, we present some recognition errors in ASR models. Due to the inherent gap between ASR models and humans in speech recognition, it is challenging to get $\bar{c}_o$ that satisfies equation (1), even for speech that contains equivalent linguistic information. Therefore, using the strict linguistic equivalence condition in equation (1) to evaluate the quality of synthesized speech can lead to inappropriate data rejection and cause a shift in the dataset distribution. Inspired by the work on Semantic Textual Similarity (Muennighoff et al., 2023), we use the similarity between $\bar{c}_o$ and $c_o$ as a criterion for judging their linguistic equivalence. Given the similarity calculation method $F$, our goal is to get $s$ to maximize the number of

$$q = F(\bar{c}, c). \tag{2}$$

where $c$ is the original text and $\bar{c}$ is the linguistic information in $s$ that usually to be the best speech recognition result by ASR model.

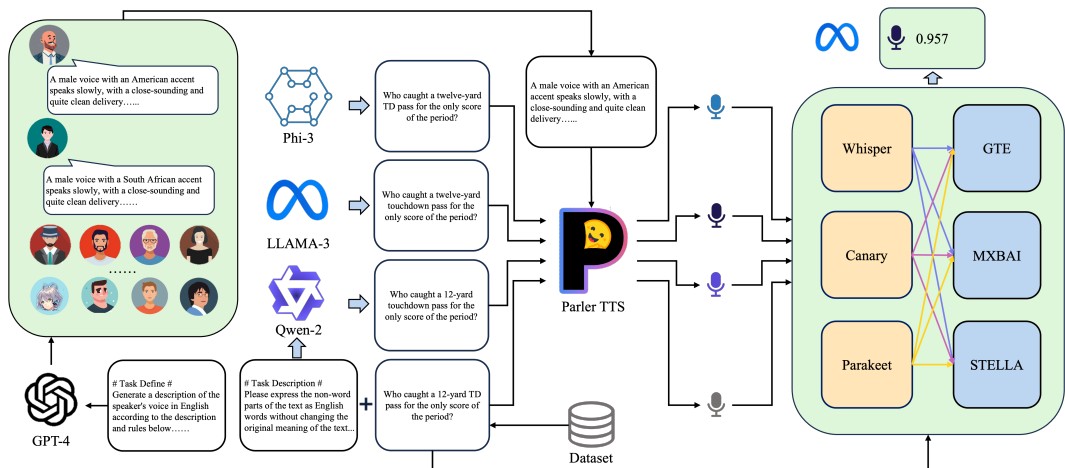

Figure 2: The structure of Question Rewriting with Multi-LLM.

## 4 METHOD

### 4.1 MULTI-AGENT ANNOTATION AND VERIFICATION

Accurate recognition of linguistic information in speech is fundamental for assessing the quality of speech instruction synthesis, as incorrect recognition can lead to improper filtering of the data. However, achieving accurate assessment is challenging due to the common recognition errors present in ASR models (Radford et al., 2022; Srivastav et al., 2023). Some human-annotated datasets integrate the opinions of multiple annotators to achieve high-quality annotation results (Deitke et al., 2024; Liu et al., 2021). Inspired by this, we propose a Multi-agent annotation and verification method. By using three ASR models and three Embedding models, we simulate multiple annotators and validators to jointly enhance the quality of the evaluation.

Specifically, for an original text instruction $c_o$ and its' synthesized speech $s_o$, we can obtain multiple recognition results $\bar{C}_o = \{ \bar{c}_{o,j} | j \in \{0, 1, 2\} \}$ through multiple ASR models $A = \{A_0, A_1, A_2\}$. Then we use embedding models $E = \{E_0, E_1, E_2\}$ to get the similarity between the original text instruction and the recognition result by

$$F(c_o, \bar{c}_{o,j}) = \frac{1}{3} \sum_{z=0}^{2} \frac{E_z(c_o) \cdot E_z(\bar{c}_{o,j})}{\|E_z(c_o)\|\|E_z(\bar{c}_{o,j})\|}. \qquad (3)$$

where $c_{o,j}^-$ is the recognition result of $s_o$ by $A_j \in A$. The quality for $s_o$ could get by

$$q(A, E) = \max_j (F(c_o, \bar{c}_{o,j})). \qquad (4)$$

When using the same similarity calculation method, higher orthogonality in ASR model performance helps to avoid consistent ASR errors, thereby improving the accuracy of the evaluation. Therefore, we use whisper-large-v3 [1] (Radford et al., 2022), canary-1b [2] and parakeet-tdt-1.1b [3] as ASR models, which have similar ASR performance in OpenASR Leaderboard (Srivastav et al., 2023) but different architectures.

### 4.2 QUESTION REWRITING FOR LINGUISTIC PRESERVATION

Due to the limited vocabulary of the TTS model, it is unable to properly convert out-of-distribution text, such as compound words, abbreviations, and mathematical formulas into speech, resulting in the loss of linguistic information. Previous work mainly relied on manually designed rules to

---

[1] https://huggingface.co/openai/whisper-large-v3
[2] https://huggingface.co/nvidia/canary-1b
[3] https://huggingface.co/nvidia/parakeet-tdt-1.1b

rewrite these contents, which inevitably led to a dependency on manual efforts, making it difficult to construct large-scale speech instruction datasets (Yang et al., 2024b). The strong performance of large language models on zero-shot tasks makes it a natural idea to use this capability for rewriting text as a solution to this problem.

Specifically, we propose a query rewriting framework based on multiple LLMs to avoid the loss of linguistic information during speech synthesis, as shown in Figure 2. For an original text instruction $c_o$, we use Llama-3-8B-Instruct (Dubey et al., 2024), Phi-3-small-8k-instruct (Abdin et al., 2024) and Qwen2-7B-Instruct (Yang et al., 2024a) to rewrite the instructions, resulting in the candidate text set $C = \{c_o, c_l, c_p, c_q\}$. To enhance the diversity of speech styles, we used GPT-4 to generate descriptions for 192 different speakers $D = \{d_0, d_1, \ldots, d_{191}\}$ to control the speech styles. Then, we can get the candidate speech set $S = \{s_o, s_l, s_p, s_q\}$ by TTS model and randomly selected description text $d \in D$. Following the evaluation and validation methods mentioned in Section 4.1, we can obtain the quality of every synthesized speech through

$$q(A, E|c, c_o) = \max_j (F(c_o, \bar{c}_j)), c \in C. \tag{5}$$

where $\bar{c}_j$ is the speech recognition result of $s \in S$ by ASR model $A_j \in A$ (4.1). Then we can obtain the optimal synthesized speech $s$ and the text $\hat{c} \in C$ which is the optimal input into TTS.

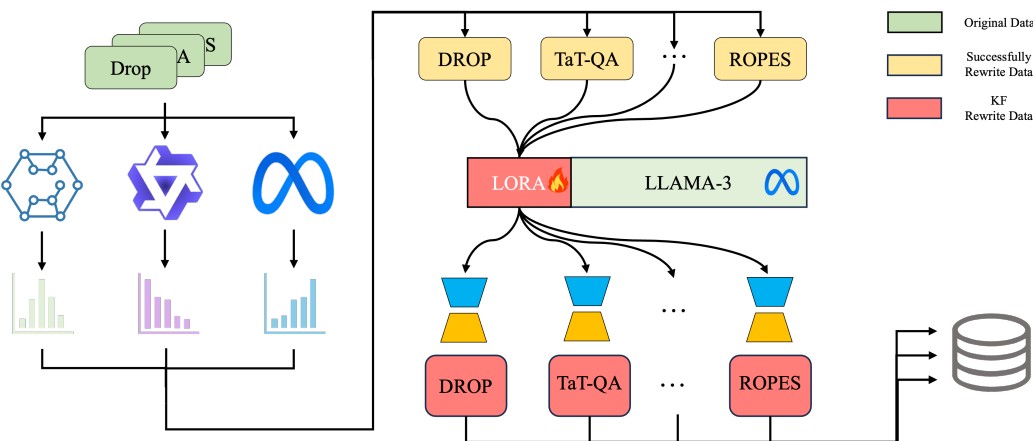

Figure 3: The structure of Knowledge Fusion.

## 4.3 KNOWLEDGE FUSION FOR CHALLENGING QUERY REWRITING

The performance of the models on challenging zero-shot tasks shows a degree of orthogonality due to differences in their knowledge. However, they exhibit limitations in tasks that require multi-perspective capabilities. Recent studies have shown that knowledge fusion can effectively leverage the knowledge of multiple models (Wan et al., 2024). By learning from data generated by different models with unique perspectives, the model's ability to understand complex tasks is significantly enhanced. Inspired by this, we propose a knowledge fusion method to address challenging rewriting tasks as shown in Figure 3. By integrating the rewriting capabilities of multiple LLMs, we aim to correct failed samples.

Specifically, for a dataset $X = \{c_0, c_1, c_2, \ldots, c_{n-1}\}$, we can obtain the optimal set $X_b = \{\hat{c}_0, \hat{c}_1, \hat{c}_2, \ldots \hat{c}_{n-1}\}$ for input into TTS using the method from Section 4.2. Then, we consider the sample pairs $< c_i, \hat{c}_i >, i \in [0, n-1]$ that satisfy $q(A, E|\hat{c}_i, c_i) > \alpha$ and $q(A, E|c_i, c_i) < \alpha$ as successfully rewritten samples for knowledge fusion training, and the sample pairs $< c_i, \hat{c}_i >, i \in [0, n-1]$ that satisfy $q(A, E|\hat{c}_i, c_i) < \alpha$ as the samples with failed rewrites, $\alpha$ is the hyperparameters used to control data quality, where $q(A, E|c_i, c_i)$ fellow equation (5). In this paper, we set $\alpha = 0.9$.

We use Meta-Llama-3-8B-Instruct (Dubey et al., 2024) as the backbone model and employ LoRA (Hu et al., 2021) to train the model. Our goal is to minimize the

$$\mathcal{L} = -\sum_{i=0}^{M} logP(y_i|x, c, y_{<i}). \tag{6}$$

where $< c, y >$ is the successfully rewritten sample, $x$ is the prompt. Then we can get the new rewrite set $C_n = \{c_{i,n} | q(A, E | \hat{c}_i, c_i) < \alpha\}$ for the samples with failed rewrites by the model with knowledge fusion. Finally, following equation (5), we can obtain the new optimal set $X_{nb}$ for input into TTS.

## 5 EXPERIMENT AND RESULT

### 5.1 EXPERIMENT SETTINGS

**Dataset** In real user interaction scenarios, it is rare to use lengthy speech to provide a detailed task definition; instead, users often ask brief questions expecting responses that meet their expectations. Therefore, in this paper, we selected several QA datasets with short questions to validate the effectiveness of our method. Specifically, we use DROP (Dua et al., 2019), Quoref (Dasigi et al., 2019), ROPES(Lin et al., 2019), NarrativeQA (Kočiský et al., 2017), TAT-QA (Zhu et al., 2021), SQUAD1.1 (Rajpurkar et al., 2016) and SQUAD2.0 (Rajpurkar et al., 2018) to validate the effectiveness of the proposed method, We provide more information about the dataset in Appendix B.3.

For the training data to train LSLMs, we calculate the data quality using Equation (5) and set a threshold $t$. The quality of all data used for training the model must exceed $t$. For data derived from the same original text, only the speech instruction with the highest quality that meets the threshold requirement is included in the training. We train LSLMs using data synthesized under the Multi-Speaker Setting, combining all datasets to train it.

**Baseline** We use the following setup as our baseline: (1) **Original**: Directly using the original text as TTS input for speech synthesis; (2) **TN** (Text Normalization): This is a commonly used method in the industry to optimize TTS inputs. We implement text normalization using the method proposed by (piAI, 2017), which achieved an accuracy of 98.27% in Text Normalization Challenge (Howard et al., 2017).

To evaluate the effectiveness of each component of our proposed method, we adopt the following configurations as ablation methods: (1) **Phi3/Qwen2/Llama3**: Follow the synthesis framework in Figure 2 but only use Phi-3-small-8k-instruct/Qwen2-7B-Instruct/Llama-3-8B-Instruct to rewrite queries; (2) **Ours w/o KF**: Use only the synthesis framework in Figure 2 without knowledge fusion.

Considering that speech style descriptions can impact speech synthesis, we adopt the following two configurations to evaluate the generalization ability of our proposed method across different TTS models: (1) **Multi-Speaker Setting**: Following the framework in Figure 2, we use GPT-4 (OpenAI et al., 2024) to generate diverse speech descriptions and employ Parler-TTS-Large-v1 as the TTS model; (2) **Single-Speaker Setting**: We include two additional widely-used vocoder-based TTS models, MeloTTS (Zhao et al., 2023) and MMS-TTS-ENG (Pratap et al., 2023), to evaluate the effectiveness of our method across three TTS models.

For the training of LSLMs, we use Qwen2-Audio-7B-Instruct as the backbone, we adopt the following two alignment target settings: (1) **Golden**: Using high-quality human-annotated answers. The dataset used in this paper includes high-quality official annotations for responses, which we have reused; (2) **Continue**: Aligning with answers generated by LLM. In this paper, we use Llama-3-8B-Instruct to generate the answers. For the test dataset, We use the MeloTTS to synthesize speech and discard all data with inconsistent linguistic information. We provide an introduction to the training method in Appendix B.2.

**Speech style control** For Multi-Speaker Setting, to enhance the diversity of speech styles and make the dataset's style distribution more closely resemble that of human datasets, following the approach of Lacombe et al. (2024), we used six attributes to describe the vocal characteristics and employed GPT-4 to generate natural language descriptions, the prompt is in Appendix A. Table 1 provides examples of speech style descriptions, with additional examples available in the appendix D. During the speech synthesis process, we randomly selected a speech description for each text. To maintain consistency, the rewritten text and the original text used the same vocal style description.

Table 1: Examples of speech style descriptions generated by GPT-4 (OpenAI et al., 2024).

| Name | Description |
|---|---|
| Luminous | A male voice with an American accent speaks slowly, enunciating each word clearly. The speaker's voice is close-sounding and quite clean, maintaining a monotone pitch throughout. The recording captures his voice with good clarity. |
| Timothy | A male voice with an American accent speaks slowly, with a close-sounding and quite clean delivery. The speaker's pitch is very monotone, and the recording captures his voice with good clarity. |
| Jocelyn | A female voice with a Canadian accent speaks slowly. The speaker's voice is close-sounding and quite clean, with a monotone pitch. The recording captures the voice with a clear and precise quality. |
| Nadine | A female voice with an American accent speaks normally. The voice is close-sounding and quite clean, with a slightly expressive and animated pitch. The recording captures a subtly engaging tone. |

Table 2: Evaluation results on SIM for different datasets under the **Multi-Speaker Setting**.

| Method | DROP | Quoref | ROPES | NarrativeQA | TAT-QA | SQUAD1.1 | SQUAD2.0 | Average |
|---|---|---|---|---|---|---|---|---|
| Original | 93.71 | 96.25 | 97.07 | 95.87 | 82.24 | 93.40 | 93.42 | 93.14 |
| TN | 95.95 | 97.18 | 97.63 | 97.75 | 89.07 | 94.88 | 94.93 | 95.34 |
| Phi3 | 97.24 | 98.07 | 98.32 | 97.64 | 95.29 | 96.39 | 96.39 | 97.05 |
| Qwen2 | 96.45 | 97.64 | 98.05 | 97.08 | 90.31 | 95.68 | 95.69 | 95.84 |
| Llama3 | 97.30 | 97.88 | 98.25 | 97.22 | 95.27 | 96.35 | 96.34 | 96.94 |
| Ours w/o KF | 98.02 | 98.57 | 98.79 | 98.14 | 97.12 | 97.37 | 97.36 | 97.91 |
| Ours | **98.11** | **98.62** | **98.82** | **98.24** | **97.18** | **97.47** | **97.49** | **97.99** |

For Single-Speaker Setting, each TTS model uses a fixed speech style. For Parler-TTS-Large-v1, we use *Jon's voice is monotone yet slightly fast in delivery, with a very close recording that almost has no background noise* as the speaker style descriptions. For MeloTTS (Zhao et al., 2023), we use the officially provided *EM-US* setting. For MMS-TTS-ENG, no additional speaker style control measures are provided by the official implementation, so we follow the default settings.

**Implementation Details**    We provide implementation details in Appendix B.1.

## 5.2 EVALUATION METRICS

**Synthetic data quality evaluation**    We use the following metrics to evaluate the performance of our proposed method in terms of data synthesis quality: (1) **SIM**: The similarity in the embedding space between the linguistic information in the speech and the original text, calculated following the method we proposed in Section 4.1; (2) **WER** (Word Error Rate): This metric is commonly used to assess the accuracy of speech recognition and is similar to the strict linguistic equivalence judgment in Equation (1); (3) **Pass**: The proportion of speech in the dataset with a quality higher than $\alpha = 0.9$, calculated according to Equation (5).

**Generative evaluation**    To evaluate the quality of the generated results in DROP, Quoref, ROPES and NarrativeQA, we use ROUGE-L (Lin, 2004) as evaluation metrics.

Table 3: Evaluation results on Pass for different datasets under the **Multi-Speaker Setting**.

| Method | DROP | Quoref | ROPES | NarrativeQA | TAT-QA | SQUAD1.1 | SQUAD2.0 | Average |
|---|---|---|---|---|---|---|---|---|
| Original | 74.40 | 84.78 | 89.80 | 83.63 | 25.85 | 73.39 | 73.45 | 72.19 |
| TN | 84.94 | 89.29 | 91.18 | 93.29 | 50.49 | 82.66 | 82.52 | 82.05 |
| Phi3 | 89.82 | 92.69 | 94.60 | 91.04 | 79.46 | 85.91 | 85.84 | 88.48 |
| Qwen2 | 87.28 | 91.70 | 94.09 | 89.38 | 56.63 | 83.46 | 83.53 | 83.72 |
| Llama3 | 90.40 | 92.23 | 94.44 | 89.37 | 80.21 | 85.92 | 85.96 | 88.36 |
| Ours w/o KF | 93.43 | 95.24 | 96.32 | 93.54 | 88.72 | 90.20 | 90.19 | 92.52 |
| Ours | **94.11** | **95.59** | **96.71** | **94.29** | **89.12** | **90.78** | **90.93** | **93.07** |

Table 4: Evaluation results of the generation quality of LSLMs trained with different methods.

| Backbone Model | Training Target | Threshold ($t$) | Data Construction Method | Drop | Quoref | Ropes | NarrativeQA | Average |
|---|---|---|---|---|---|---|---|---|
| | - | - | - | 17.40 | 55.98 | 42.69 | 43.02 | 39.77 |
| | Golden | 0.00 | Original | 29.25 | 76.01 | 55.42 | 48.34 | 52.26 |
| | LLM Continue | 0.00 | Ours | 30.08 | 75.05 | 57.15 | 47.88 | 52.54 |
| Qwen2-Audio-7B-Instruct | Golden | 0.00 | Ours | 42.78 | 86.58 | 60.48 | 52.15 | 60.50 |
| | Golden | 85.00 | Ours | 42.95 | 85.18 | 56.47 | 53.61 | 59.55 |
| | Golden | 90.00 | Ours | **44.35** | **86.81** | **64.24** | **56.76** | **68.43** |
| | Golden | 95.00 | Ours | 41.86 | 86.73 | 58.55 | 54.61 | 60.44 |

Table 5: Evaluation results of different ASR methods under the Multi-Speaker Setting. Using WER as the evaluation metric.

| ASR Method | DROP | Quoref | ROPES | NarrativeQA | TAT-QA | SQUAD1.1 | SQUAD2.0 | Average |
|---|---|---|---|---|---|---|---|---|
| canary | 10.93 | 5.46 | 6.97 | 8.88 | 18.61 | 10.19 | 9.55 | 10.08 |
| whisper | 11.18 | 5.32 | 7.09 | 8.53 | 19.98 | 9.67 | 9.71 | 10.21 |
| parakeet | 10.42 | 5.08 | 6.21 | 8.39 | 18.44 | 9.79 | 9.14 | 9.64 |
| Ours | **9.19** | **4.14** | **5.18** | **6.21** | **18.31** | **7.74** | **7.74** | **8.36** |

## 5.3 MAIN RESULT

**Synthetic data quality** For the Multi-Speaker Setting, we present the evaluation results of the SIM and Pass in Table 2 and Table 3. The experimental results demonstrate that our proposed method consistently shows effectiveness across all datasets, increasing the similarity between the linguistic information of the synthetic speech and the original text in the embedding space from 93.06% to 97.98%. For the Single-Speaker Setting, we present the evaluation results of the SIM and Pass in Table 14 and Table 15. The experimental results on multiple TTS models demonstrate the excellent generalization ability of our proposed method. Using our method, the absolute difference in embedding space quality between MeloTTS and Parler-TTS-Large-v1 is reduced from 3.96% to 0.09%, and the gap in data usability is narrowed from 13.95% to 0.9%. This bridges the quality gap in synthesized data between vocoder-based TTS models and autoregressive TTS models.

**Use Synthetic data finetune LSLMs** As shown in Table 4, we present the evaluation results of models trained under different experimental settings. The experimental results demonstrate that using *Golden* as the alignment target exhibits significant superiority compared to the *Continue* approach. This provides new insights into the training of LSLMs, indicating that in the process of aligning speech instructions, continuation data generated by LLMs cannot replace high-quality human-annotated data. On the other hand, training the model with training sets obtained using different sampling thresholds achieved the best performance at a threshold of 0.90. This validates the rationality of our threshold setting in the *Pass* metric. It also demonstrates that discarding low-quality samples through an appropriate threshold can effectively improve the model's performance.

## 5.4 ABLATION EXPERIMENT

**Orthogonality of LLM performance** As mentioned in Section 4.2, the degree of orthogonality among different LLMs in this zero-shot task is positively correlated with the improvement in quality. We present more detailed evaluation results across multiple datasets in Tables 2 and 14. The experimental results show that the performance of using a single LLM is inferior to that of using multiple LLMs together, and there are subtle differences in the performance of different LLMs on this task.

**The effectiveness of multi-agent annotation and validation** As shown in Tables 5, we present the evaluation results of using different ASR models for speech recognition. The experimental results demonstrate that the combined use of multiple different ASR models consistently improves WER, showcasing the advantages of this approach in reducing automatic annotation errors and avoiding inappropriate data filtering. Meanwhile, additional evaluation results under various similarity calculation methods are provided in Appendix C.2, consistently verifying the effectiveness of the approach. As shown in Table 6, we provide the performance of different similarity calculation methods in selecting recognition results based on synthesized speech from original text. We use

Table 6: Results using different word embedding models on various datasets, without LLM-based rewriting. All datasets are evaluated using WER as the metric.

| Embedding Model | DROP↓ | ROPES↓ | Ropes↓ | NarrativeQA↓ | Tat-Qa↓ | SQUAD1.1↓ | SQUAD2.0↓ | Average↓ |
|---|---|---|---|---|---|---|---|---|
| gte | 9.289 | 4.144 | 5.193 | 6.296 | 18.980 | 7.857 | 7.869 | 8.518 |
| mxbai | 9.256 | 4.170 | 5.205 | 6.221 | 18.331 | 7.792 | 7.786 | 8.394 |
| stella | **9.129** | 4.211 | 5.326 | 6.330 | **18.212** | **7.765** | 7.768 | 8.392 |
| gte+mxbai+stella | 9.188 | **4.143** | **5.176** | **6.209** | 18.312 | 7.741 | **7.738** | **8.358** |

Table 7: Estimated results of the experimental cost.

| Speech Collection Method | Quality Validation | Speakers Num | Human Time Cost | Gpu Time Cost | Money Cost |
|---|---|---|---|---|---|
| Human | Human | ∞ | 562 | 0 | 4215 |
| Human | Single ASR | ∞ | 281 | 6 | 2110.02 |
| Human | Multi ASR + Emb | ∞ | 281 | 16 | 2114.22 |
| MeloTTS+Original | Single ASR | 1 | 0 | 14 | 5.88 |
| Parler+Original | Single ASR | 192 | 0 | 127 | 53.34 |
| MeloTTS+Ours w/o KF | Multi ASR + Emb | 1 | 0 | 82 | 34.44 |
| Parler+Ours w/o KF | Multi ASR + Emb | 192 | 0 | 534 | 224.28 |
| Parler+Ours | Multi ASR + Emb | 192 | 0 | 558 | 234.36 |

WER as the evaluation metric here, primarily because the original text was directly used for synthesizing speech, without altering the text distribution. WER is more suitable than semantic similarity for assessing data quality in this context. The experiments show that using the average semantic similarity of multiple embedding models, compared to a single model, offers certain advantages in terms of average WER across various datasets.

## 5.5 EXPERIMENTAL COST COMPARISON

To thoroughly demonstrate the advantages of our proposed method in terms of efficiency and experimental costs, we present the experimental expenses calculated based on the lowest standard, as shown in Table 7. For labor costs, we calculate the time for speech collection and data annotation at a 1:1 ratio, using a labor rate of $7.50 per hour, which is the minimum wage mandated by the USA government. In reality, the minimum wage across states is generally higher than this. Meanwhile, we calculate GPU costs using the lowest rate for a single NVIDIA A40 GPU card from cloud service providers, which is $0.42 per hour per device, excluding any additional time costs related to data transfer and other operations. The experimental results show that the cost of our method is less than one-tenth of that of high-quality human datasets, achieving a good balance between expense and dataset quality.

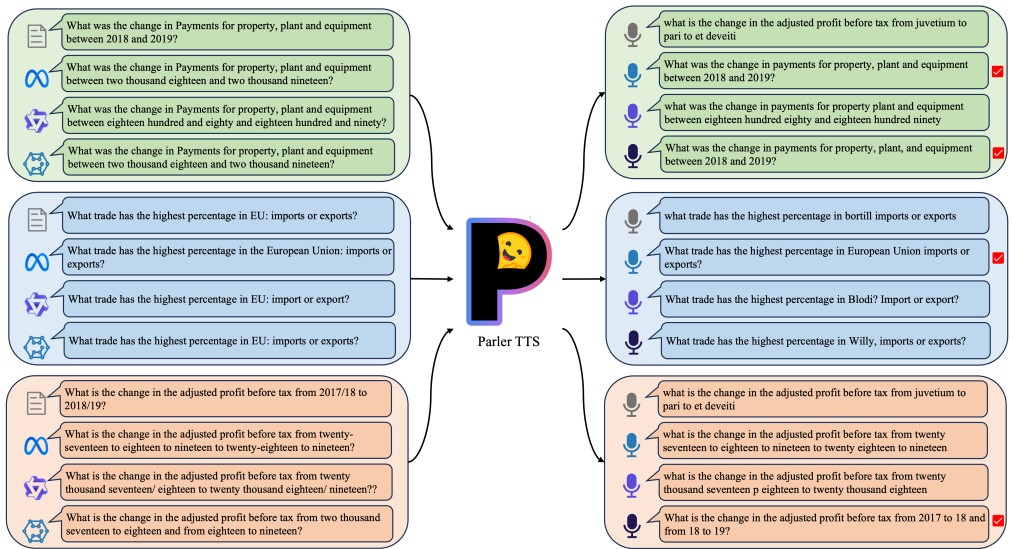

Figure 4: Examples of successfully rewritten queries.

## 5.6 CASE STUDY

As shown in Figure 4, we present some examples of successful rewrites. We demonstrate the unique advantages of our proposed method through three typical rewriting scenarios: (1) Rule-based rewriting, (2) Context-aware rewriting, and (3) Rewriting requiring complex comprehension abilities. In the Rule-based rewriting example, the LLM follows the rule of converting numbers into English words. In the Context-aware rewriting scenario, Llama-3-8B-Instruct combines information from the context to infer that *EU* is the abbreviation for European Union and correctly completes the rewrite. In the Rewriting requiring complex comprehension abilities, Phi-3-small-8k-instruct successfully combines context and inference to deduce that *2019/18* in the original text refers to the period from 2017 to 2018, and rewrites the text into a form more suitable for speech synthesis. These examples illustrate the unique advantages of our proposed method in both adhering to established rules and leveraging comprehension to tackle complex rewrites.

As shown in Table 8, we provide response examples from models trained with different speech data and alignment objectives. The results indicate that the model trained with high-quality speech data aligned with human responses accurately perceived the speech instructions and produced results that closely matched human responses.

Table 8: Example of responses from different models.

| Context | |
|---|---|
| System Prompt | This is a chat between a user and an artificial intelligence assistant. The assistant gives helpful, detailed, and polite answers to the user's questions based on the context. The assistant should also indicate when the answer cannot be found in the context. |
| Document | The story follows a dinner party given by Bertha Young and her husband, Harry. The writing shows Bertha depicted as a happy soul, though quite naive about the world she lives in and those closest to her. The story opened up a lot of questions, about deceit, about knowing oneself and also about the possibility of homosexuality at the start of the 20th century. The story gives us a bird's eye view of the dinner party, which is attended by a couple, Mr. and Mrs. Norman Knight, who are close friends to Bertha and Harry. Guest, Eddie Warren, is an effeminate character, who adds an interesting mix to the party. The only other guest, Pearl Fulton, is someone who Bertha is mysteriously drawn to for reasons unknown to her at the start. The interesting thing is that Bertha's husband is presented to the reader as Bertha perceives him in her mind. Because Bertha is so naive, the reader first gets the impression that Harry is a crude, disinterested person who has a strong dislike for Pearl by his conversational tone and curtness towards her as the conversation unfolds. As the dinner party progresses, Bertha questions her own interest and fascination towards Pearl. The fact that Eddie, who is most likely homosexual, is present, lends an air to the possibility that Bertha's interest in Pearl is more than a platonic feeling one has towards a friend of the same sex. It is only after Bertha analyzes her feelings towards Pearl that she realizes that the connection she feels with Pearl is their mutual attraction for Harry, and coming out of her "blissful" reverie she makes the discovery that Harry and Pearl are having an affair. The title to this story alludes to the sentiment that ignorance is bliss. The story leaves the question about whether it is best to live blissfully ignorant of the truth or live with the knowledge of a harsh reality. |
| Prompt | Answer the following question with a short span. The answer needs to be just in a few words. What is Bertha's downfall when it comes to observing life and people? |
| Speech Instruction | What is Bertha's downfall when it comes to observing life and people? |
| Response | |
| Reference | She is naive. |
| Original+Golden | Harry. |
| Ours+Continue | Bertha's downfall comes from her naivety and inability to see the truth. |
| Ours+Golden | Bertha is naive. |

## 6 CONCLUSION

In this paper, we propose a query rewriting method based on multi-LLM knowledge fusion, with multi-agent annotation and validation for data quality. Experiments demonstrate that this method consistently performs well across multiple datasets, improving the average data usability from 72% to 93%. Through ablation studies, we analyzed the effectiveness of each component, and the results show that different LLMs exhibit a certain degree of orthogonality in this zero-shot task. Moreover, using multiple annotation agents helps to better reduce data quality evaluation errors and improper data filtering caused by recognition and annotation errors of a single model. Our work enables the automated construction of high-quality language instruction datasets.

REPRODUCIBILITY STATEMENT

**Implementation details**   We give the Implementation details in Section 5.1.

**Code And Dataset**   We provide the main code of this paper in the supplementary materials, along with some data examples. The complete project files will be compiled in the near future and open sourced in the github repository after the paper is accepted.

**Assets and licenses**   We have provided assets and licenses on all the open-source datasets, open-source models, and key open-source project used in this paper in Appendix E, along with download URL.

ETHICS STATEMENT

This study adheres to relevant ethical standards. The research team is committed to ensuring thetransparency and reproducibility of the code while taking measures to avoid potential discriminationand bias. The findings of this study aim to advance scientific understanding while ensuring noharmful impacts on society.

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

## A  THE PROMPTS USED

Table 9: Prompts input to GPT-4 for generating speech descriptions.

---

# Task Define #
Generate a description of the speaker's voice in English according to the description and rules below.

# The rules #
(1) Six characteristics are given below to describe the speaker's voice
(2) Please refer to the features and examples to give corresponding descriptions

# Example #

A female voice with an American accent enunciates every word with precision. The speaker's voice is very close-sounding and clean, and the recording is excellent, capturing her voice with crisp clarity.

---

Table 10: Prompt for query rewriting.

---

# Task Define # Please express the non-word parts of the text as English words without changing the original meaning of the text. Follow the format of the example and output the result directly without any output that is not related to the result.

# The rules #

(1) For the year, month and other parts containing numbers, please use English words to express these numbers.

(2) For Roman numerals and Greek symbols. Please convert the same form to the corresponding English word.

(3) For the symbols of chemistry, physics and other fields, please express these symbols in the form of English words.

---

Table 11: The prompts used for fine-tuning LSLMs. *Document* refers to the text associated with the *Speech*, *Speech* represents the user's query, and *Response* refers to the reference reply, which serves as the alignment target for training.

```
<|im_start|>system

This is a chat between a user and an artificial intelligence assistant. The assistant gives help-
ful, detailed, and polite answers to the user's questions based on the context. The assistant
should also indicate when the answer cannot be found in the context.

{document}
<|im_end|>
<|im_start|>user
Answer the following question with a short span. The answer needs to be just in a few
words.
{Speech} <|im_end|>
<|im_start|>assistant
{Response}
<|im_end|>
```

Table 12: Prompts for consistency verification.

```
# Task #

Please evaluate whether the following two sentences convey identical meanings in content.
Use yes or no to give your verdict.

# Sentence 1 #

{original_text}

# Sentence 2 #

{rewrite_text}
```

# B MORE EXPERIMENT SETTINGS

## B.1 IMPLEMENTATION DETAILS

In this paper, we use LoRA (Hu et al., 2021) for training. For the training in Knowledge Fusion, we setting $r = 8$ and $a = 16$. The peak learning rate was set to 3e-4, and the cosine scheduler was used for learning rate adjustment. For the training in LSLMs Finetune, we set $r = 8$ and $a = 32$. The peak learning rate was set to 3e-5, and the cosine scheduler was used for learning rate adjustment. During training, we utilized 4 NVIDIA A40 GPUs, enabled gradient checkpointing to save memory, and applied gradient accumulation with backward occurring every 64 samples. The batch size per device was set to 1. For the generative evaluation phase, we evaluated our results on a single NVIDIA A40 GPU, setting top-p to 0.5, top-k to 20, repetition penalty to 1.1, and temperature to 0.7. For speech synthesis, we use a single NVIDIA A40 GPU with the batch size set to 1.

## B.2 TRAINING METHOD FOR LSLMS

We follow the method of (Chu et al., 2023) to train LSLMs. Specifically, for a speech instruction $s$ and its text response $y = (y_1, y_2, \ldots, y_T)$, our goal is to maximize the product of the conditional probabilities:

$$P(y|d, s) = \prod_{t=1}^{T} P(y_t|d, \text{Encoder}(s), y_{1:t-1})$$

where d represents the text context, Encoder represents the audio encoder in LSLMs.

## B.3 MORE INFORMATION ABOUT THE DATASET

In Table 13, we provide more information about the dataset we used.

Table 13: Description and num of the datasets used in this paper. *Num* refers to the number of samples in the dataset.

| Dataset Name | Description | Num |
|---|---|---|
| DROP | A reading comprehension dataset that requires systems to perform discrete reasoning. | 29k |
| Quoref | A reading comprehension dataset that focuses on coreference resolution. | 11k |
| ROPES | A reading comprehension dataset requires reasoning about the effects of causes in unfamiliar situations using provided passages. | 10k |
| NarrativeQA | A reading comprehension dataset requires answering questions based on understanding long, complex narrative texts. | 40k |
| TAT-QA | A tabular and textual question-answering dataset requiring numerical reasoning over tables and text. | 11k |
| SQUAD1.1 | A reading comprehension dataset where models answer questions based on Wikipedia passages. | 87k |
| SQUAD2.0 | It extends SQuAD1.1 by adding unanswerable questions, requiring models to not only answer questions based on Wikipedia passages but also determine when no answer is possible. | 87k |

## B.4 THE EMBEDDING MODELS USED IN THIS PAPER

**gte-large-en-v1.5** is a text embedding model from the GTE-v1.5 series, developed by the Institute for Intelligent Computing at Alibaba Group. The model supports a context length of up to 8192 and is built on a Transformer++ encoder backbone, which combines BERT, RoPE, and GLU technologies.

**mxbai-embed-large-v1** is a text embedding model which developed by Mixedbread.

**stella_en_400M_v5** is a text embedding model that is based on the Alibaba-NLP/gte-large-en-v1.5 and Alibaba-NLP/gte-Qwen2-1.5B-instruct models. The model simplifies the use of prompts, providing two main prompts for general tasks: one for sentence-to-paragraph (s2p) and another for sentence-to-sentence (s2s) tasks.

# C MORE EXPERIMENTAL RESULTS

## C.1 EXPERIMENTAL RESULTS UNDER THE SINGLE-SPEAKER SETTING

Table 14: Evaluation results on SIM for different datasets under the Single-Speaker Setting.

| TTS Model | Method | Drop | narrativeqa | Quoref | ropes | squad1.1 | squad2.0 | tatqa | Average |
|---|---|---|---|---|---|---|---|---|---|
| MeloTTS | Original | 96.61 | 95.13 | 97.27 | 98.22 | 95.75 | 93.48 | 97.43 | 96.27 |
| | TN | 97.50 | 96.36 | 98.00 | 98.83 | 94.86 | 95.54 | 97.96 | 97.01 |
| | Phi3 | 97.57 | 95.82 | 97.81 | 98.70 | 96.74 | 95.71 | 98.29 | 97.23 |
| | Qwen2 | 97.59 | 96.42 | 98.01 | 98.85 | 96.98 | 95.68 | 98.19 | 97.39 |
| | Llama3 | 97.65 | 96.32 | 98.01 | 98.91 | 97.10 | 95.78 | 98.30 | 97.44 |
| | Ours w/o KF | **98.19** | **97.02** | **98.37** | **99.12** | **97.59** | **96.80** | **98.53** | **97.94** |
| MMS-TTS-ENG | Original | 92.05 | 91.46 | 93.77 | 93.42 | 91.28 | 91.23 | 84.61 | 91.12 |
| | TN | 94.22 | **93.82** | 95.64 | 95.14 | 93.86 | 93.26 | 90.75 | 93.81 |
| | Phi3 | 95.06 | 93.00 | 95.46 | 94.97 | 93.90 | 93.86 | 92.52 | 94.11 |
| | Qwen2 | 95.18 | 91.47 | 95.51 | 95.04 | 94.32 | 94.62 | 91.81 | 93.99 |
| | Llama3 | 94.52 | 91.47 | 95.51 | 94.98 | 94.85 | 94.82 | 94.30 | 94.35 |
| | Ours w/o KF | **96.86** | 93.01 | **96.51** | **95.95** | **95.96** | **96.07** | **96.17** | **95.79** |
| Parler-TTS-Large-v1 | Original | 93.04 | 95.26 | 95.51 | 96.01 | 92.75 | 91.99 | 81.63 | 92.31 |
| | TN | 96.41 | 97.39 | 97.63 | 98.06 | 95.89 | 95.61 | 89.07 | 95.72 |
| | Phi3 | 97.21 | 97.40 | 97.95 | 97.97 | 96.49 | 96.19 | 93.80 | 96.72 |
| | Qwen2 | 97.17 | 97.51 | 97.93 | 98.05 | 96.29 | 95.64 | 91.63 | 96.32 |
| | Llama3 | 97.41 | 97.41 | 97.78 | 96.74 | 96.74 | 96.44 | 96.01 | 96.93 |
| | Ours w/o KF | **98.09** | **98.19** | **98.52** | **98.30** | **97.31** | **97.32** | **97.21** | **97.85** |

Table 15: Evaluation results on Pass for different datasets under the Single-Speaker Setting.

| TTS Model | Method | Drop | narrativeqa | Quoref | ropes | squad1.1 | squad2.0 | tatqa | Average |
|---|---|---|---|---|---|---|---|---|---|
| MeloTTS | Original | 86.12 | 79.21 | 88.57 | 93.21 | 82.91 | 74.87 | 90.25 | 85.02 |
| | TN | 90.04 | 84.60 | 92.19 | 95.84 | 83.53 | 82.38 | 92.46 | 88.72 |
| | Phi3 | 90.51 | 82.16 | 91.08 | 95.41 | 87.20 | 83.30 | 93.79 | 89.07 |
| | Qwen2 | 90.59 | 85.27 | 92.20 | 96.17 | 88.32 | 83.32 | 93.42 | 89.90 |
| | Llama3 | 90.84 | 84.63 | 92.12 | 96.48 | 88.74 | 83.58 | 93.93 | 90.05 |
| | Ours w/o KF | **93.27** | **87.89** | **93.96** | **97.37** | **90.86** | **87.69** | **94.70** | **92.25** |
| MMS-TTS-ENG | Original | 64.55 | 63.18 | 72.36 | 73.53 | 64.58 | 64.28 | 31.22 | 61.96 |
| | TN | 76.63 | **72.99** | 81.00 | 81.84 | 74.56 | 71.95 | 59.02 | 74.00 |
| | Phi3 | 80.50 | 69.30 | 80.37 | 81.11 | 75.01 | 74.80 | 66.08 | 75.31 |
| | Qwen2 | 81.38 | 63.24 | 80.83 | 81.51 | 77.14 | 78.34 | 61.32 | 74.82 |
| | Llama3 | 81.48 | 63.18 | 80.56 | 81.24 | 79.13 | 78.97 | 73.09 | 76.81 |
| | Ours w/o KF | **86.98** | 69.33 | **85.76** | **86.49** | **84.08** | **96.07** | **84.19** | **84.70** |
| Parler-TTS-Large-v1 | Original | 73.56 | 82.91 | 83.68 | 88.04 | 72.83 | 70.41 | 26.07 | 71.07 |
| | TN | 85.52 | 90.54 | 91.05 | 94.44 | 83.13 | 82.43 | 49.38 | 82.36 |
| | Phi3 | 90.25 | 90.47 | 92.92 | 94.10 | 86.74 | 85.91 | 75.21 | 87.94 |
| | Qwen2 | 90.40 | 91.16 | 92.42 | 94.64 | 86.24 | 83.92 | 63.68 | 86.07 |
| | Llama3 | 91.34 | 90.59 | 92.16 | 93.61 | 87.88 | 87.14 | 85.22 | 89.71 |
| | Ours w/o KF | **94.37** | **94.32** | **95.61** | **96.39** | **90.37** | **90.85** | **90.12** | **93.15** |

## C.2 RESULTS EVALUATED BY DIFFERENT WORD EMBEDDING MODELS

Table 16: The quality evaluation results of our proposed method in datasets. The similarity here is the average of the results calculated by gte-large-en-v1.5 (Appendix B.4).

| Method | ASR Model | DROP | | | Quoref | | | ROPES | | |
|---|---|---|---|---|---|---|---|---|---|---|
| | | WER↓ | SIM↑ | Pass↑ | WER↓ | SIM↑ | Pass↑ | WER↓ | SIM↑ | Pass↑ |
| - | canary | 10.928 | 91.301 | 67.491 | 5.455 | 93.745 | 76.792 | 6.971 | 95.572 | 82.754 |
| - | whisper | 11.175 | 91.568 | 69.139 | 5.322 | 93.964 | 77.965 | 7.087 | 95.458 | 82.726 |
| - | parakeet | 10.416 | 90.881 | 67.717 | 5.084 | 93.632 | 77.656 | 6.208 | 95.419 | 84.237 |
| - | Ours | **9.289** | **93.488** | **74.396** | **4.144** | **95.836** | **84.058** | **5.193** | **97.122** | **89.290** |
| phi3 | Ours | 4.967 | 97.225 | 89.169 | 2.533 | 97.877 | 91.788 | 3.602 | 98.368 | 94.150 |
| llama3 | Ours | 5.458 | 97.299 | 89.772 | 3.219 | 97.683 | 91.188 | 4.071 | 98.295 | 94.077 |
| qwen2 | Ours | 7.902 | 96.633 | 87.577 | 3.884 | 97.507 | 90.806 | 5.238 | 98.161 | 93.757 |
| Ours w/o KF | Ours | **4.683** | **98.078** | **92.896** | **2.518** | **98.475** | **94.352** | **3.467** | **98.833** | **96.146** |

| Method | ASR Model | NarrativeQA | | | TAT-QA | | |
|---|---|---|---|---|---|---|---|
| | | WER↓ | SIM↑ | Pass↑ | WER↓ | SIM↑ | Pass↑ |
| - | canary | 8.880 | 93.011 | 72.518 | 18.608 | 75.349 | 21.685 |
| - | whisper | 8.527 | 93.149 | 73.500 | 19.981 | 75.993 | 22.259 |
| - | parakeet | 8.387 | 91.975 | 71.255 | 18.441 | 76.808 | 21.146 |
| - | Ours | 6.296 | 95.572 | 82.375 | 18.980 | 79.366 | 24.633 |
| phi3 | Ours | 4.326 | 97.432 | 89.725 | 7.784 | 94.583 | 77.193 |
| llama3 | Ours | 5.637 | 97.017 | 88.140 | 9.398 | 94.593 | 78.263 |
| qwen2 | Ours | 6.532 | 97.008 | 88.593 | 17.817 | 89.507 | 52.456 |
| Ours w/o KF | Ours | 4.451 | 98.035 | 92.527 | 6.894 | 96.675 | 86.923 |

| Rewrite LLM | ASR Model | SQUAD1.1 | | | SQUAD2.0 | | |
|---|---|---|---|---|---|---|---|
| | | WER↓ | SIM↑ | Pass↑ | WER↓ | SIM↑ | Pass↑ |
| - | canary | 10.186 | 89.936 | 64.974 | 9.553 | 90.634 | 65.544 |
| - | whisper | 9.672 | 90.760 | 66.432 | 9.711 | 90.810 | 66.478 |
| - | parakeet | 9.788 | 89.588 | 65.401 | 9.139 | 90.257 | 65.762 |
| - | Ours | 7.857 | 93.054 | 72.938 | 7.869 | 93.080 | 73.043 |
| phi3 | Ours | 5.575 | 96.189 | 85.164 | 5.532 | 96.185 | 85.101 |
| llama3 | Ours | 7.014 | 96.184 | 85.218 | 6.996 | 96.177 | 85.252 |
| qwen2 | Ours | 8.659 | 95.633 | 83.233 | 8.641 | 95.638 | 83.334 |
| Ours w/o KF | Ours | 6.051 | 97.299 | 89.662 | 6.049 | 97.284 | 89.612 |

Table 17: The quality evaluation results of our proposed method in datasets. The similarity here is the average of the results calculated by mxbai-embed-large-v1 (Appendix B.4).

| Method | ASR Model | DROP | | | Quoref | | | ROPES | | |
|---|---|---|---|---|---|---|---|---|---|---|
| | | WER↓ | SIM↑ | Pass↑ | WER↓ | SIM↑ | Pass↑ | WER↓ | SIM↑ | Pass↑ |
| - | canary | 10.928 | 90.239 | 64.929 | 5.455 | 93.874 | 76.701 | 6.971 | 95.069 | 81.930 |
| - | whisper | 11.175 | 90.490 | 66.607 | 5.322 | 94.061 | 77.583 | 7.087 | 94.796 | 81.866 |
| - | parakeet | 10.416 | 89.920 | 66.056 | 5.084 | 93.209 | 76.973 | 6.208 | 94.836 | 83.422 |
| - | Ours | 9.256 | 92.677 | 71.714 | 4.170 | 95.836 | 83.467 | 5.205 | 96.732 | 88.594 |
| phi3 | Ours | 4.977 | 96.565 | 87.032 | 2.563 | 97.782 | 91.306 | 3.653 | 98.127 | 93.867 |
| llama3 | Ours | 5.466 | 96.628 | 87.649 | 3.152 | 97.602 | 90.879 | 4.078 | 98.026 | 93.537 |
| qwen2 | Ours | 7.836 | 95.913 | 85.114 | 3.917 | 97.481 | 90.951 | 5.332 | 97.890 | 93.436 |
| Ours w/o KF | Ours | 4.756 | 97.508 | 91.012 | 2.497 | 98.387 | 94.171 | 3.480 | 98.656 | 95.817 |

| Method | ASR Model | NarrativeQA | | | TAT-QA | | |
|---|---|---|---|---|---|---|---|
| | | WER↓ | SIM↑ | Pass↑ | WER↓ | SIM↑ | Pass↑ |
| - | canary | 8.880 | 92.451 | 70.950 | 18.608 | 77.890 | 22.076 |
| - | whisper | 8.527 | 92.676 | 71.990 | 19.981 | 78.071 | 22.355 |
| - | parakeet | 8.387 | 92.030 | 71.198 | 18.441 | 79.429 | 21.972 |
| - | Ours | 6.221 | 95.256 | 81.240 | 18.331 | 81.174 | 24.954 |
| phi3 | Ours | 4.258 | 97.221 | 88.945 | 7.857 | 94.705 | 76.863 |
| llama3 | Ours | 5.474 | 96.764 | 87.197 | 9.136 | 94.636 | 76.941 |
| qwen2 | Ours | 6.353 | 96.785 | 87.873 | 17.150 | 89.297 | 54.143 |
| Ours w/o KF | Ours | 4.300 | 97.844 | 91.843 | 6.443 | 96.815 | 86.566 |

| Rewrite LLM | ASR Model | SQUAD1.1 | | | SQUAD2.0 | | |
|---|---|---|---|---|---|---|---|
| | | WER↓ | SIM↑ | Pass↑ | WER↓ | SIM↑ | Pass↑ |
| - | canary | 10.186 | 89.429 | 64.035 | 9.553 | 90.112 | 64.475 |
| - | whisper | 9.672 | 90.215 | 65.480 | 9.711 | 90.264 | 65.553 |
| - | parakeet | 9.788 | 89.013 | 64.434 | 9.139 | 89.661 | 64.881 |
| - | Ours | 7.792 | 92.632 | 71.678 | 7.786 | 92.647 | 71.726 |
| phi3 | Ours | 5.517 | 95.923 | 84.290 | 5.488 | 95.919 | 84.210 |
| llama3 | Ours | 6.910 | 95.855 | 84.071 | 6.903 | 95.847 | 84.071 |
| qwen2 | Ours | 8.494 | 95.290 | 82.023 | 8.484 | 95.300 | 82.162 |
| Ours w/o KF | Ours | 5.986 | 97.030 | 88.820 | 5.950 | 97.020 | 88.790 |

Table 18: The quality evaluation results of our proposed method in datasets. The similarity here is the average of the results calculated by stella_en_400M_v5 (Appendix B.4).

| Method | ASR Model | DROP | | | Quoref | | | ROPES | | |
|---|---|---|---|---|---|---|---|---|---|---|
| | | WER↓ | SIM↑ | Pass↑ | WER↓ | SIM↑ | Pass↑ | WER↓ | SIM↑ | Pass↑ |
| - | canary | 10.928 | 93.968 | 76.376 | 5.455 | 96.079 | 84.804 | 6.971 | 96.398 | 87.294 |
| - | whisper | 11.175 | 93.806 | 76.232 | 5.322 | 96.009 | 84.876 | 7.087 | 95.967 | 86.012 |
| - | parakeet | 10.416 | 93.251 | 75.242 | 5.084 | 95.272 | 84.895 | 6.208 | 95.545 | 87.596 |
| - | Ours | 9.129 | 95.394 | 82.028 | 4.211 | 97.311 | 90.342 | 5.326 | 97.581 | 92.411 |
| phi3 | Ours | 4.929 | 98.228 | 95.177 | 2.590 | 98.727 | 96.708 | 3.849 | 98.663 | 96.192 |
| llama3 | Ours | 5.359 | 98.259 | 95.592 | 3.139 | 98.580 | 96.490 | 4.223 | 98.623 | 96.476 |
| qwen2 | Ours | 7.426 | 97.283 | 92.201 | 3.663 | 98.230 | 95.298 | 5.356 | 98.388 | 95.798 |
| Ours w/o KF | Ours | 4.490 | 98.787 | 97.503 | 2.399 | 99.075 | 98.081 | 3.656 | 99.061 | 97.620 |

| Method | ASR Model | NarrativeQA | | | TAT-QA | | |
|---|---|---|---|---|---|---|---|
| | | WER↓ | SIM↑ | Pass↑ | WER↓ | SIM↑ | Pass↑ |
| - | canary | 8.880 | 95.257 | 82.498 | 18.608 | 85.716 | 30.345 |
| - | whisper | 8.527 | 95.180 | 82.428 | 19.981 | 85.661 | 31.241 |
| - | parakeet | 8.387 | 94.083 | 81.247 | 18.441 | 84.981 | 27.911 |
| - | Ours | 6.330 | 97.009 | 89.823 | 18.212 | 87.321 | 35.562 |
| phi3 | Ours | 4.342 | 98.461 | 95.740 | 7.528 | 97.012 | 89.740 |
| llama3 | Ours | 5.523 | 98.146 | 94.857 | 8.836 | 96.989 | 90.210 |
| qwen2 | Ours | 6.075 | 97.837 | 93.648 | 16.726 | 92.936 | 72.959 |
| Ours w/o KF | Ours | 4.230 | 98.826 | 97.343 | 6.058 | 98.214 | 94.748 |

| Rewrite LLM | ASR Model | SQUAD1.1 | | | SQUAD2.0 | | |
|---|---|---|---|---|---|---|---|
| | | WER↓ | SIM↑ | Pass↑ | WER↓ | SIM↑ | Pass↑ |
| - | canary | 10.186 | 92.488 | 72.194 | 9.553 | 93.203 | 72.817 |
| - | whisper | 9.672 | 93.012 | 72.568 | 9.711 | 93.053 | 72.555 |
| - | parakeet | 9.788 | 91.678 | 71.369 | 9.139 | 92.353 | 71.717 |
| - | Ours | 7.765 | 94.869 | 79.243 | 7.768 | 94.893 | 79.299 |
| phi3 | Ours | 5.415 | 97.372 | 90.391 | 5.387 | 97.381 | 90.379 |
| llama3 | Ours | 6.732 | 97.327 | 90.619 | 6.742 | 97.326 | 90.586 |
| qwen2 | Ours | 8.022 | 96.574 | 87.689 | 7.992 | 96.579 | 87.733 |
| Ours w/o KF | Ours | 5.676 | 98.121 | 93.862 | 5.654 | 98.107 | 93.746 |

Table 19: The quality evaluation results of our proposed method in datasets. The similarity here is the average of the results calculated by gte-large-en-v1.5 and mxbai-embed-large-v1 (Appendix B.4).

| Method | ASR Model | DROP | | | Quoref | | | ROPES | | |
|---|---|---|---|---|---|---|---|---|---|---|
| | | WER↓ | SIM↑ | Pass↑ | WER↓ | SIM↑ | Pass↑ | WER↓ | SIM↑ | Pass↑ |
| - | canary | 10.928 | 90.770 | 65.679 | 5.455 | 93.810 | 76.619 | 6.971 | 95.320 | 82.479 |
| - | whisper | 11.175 | 91.029 | 67.501 | 5.322 | 94.012 | 77.410 | 7.087 | 95.127 | 82.314 |
| - | parakeet | 10.416 | 90.401 | 66.501 | 5.084 | 93.420 | 77.155 | 6.208 | 95.127 | 84.044 |
| - | Ours | 9.254 | 92.980 | 72.540 | 4.152 | 95.777 | 83.439 | 5.157 | 96.876 | 88.887 |
| phi3 | Ours | 4.930 | 96.830 | 87.905 | 2.513 | 97.785 | 91.324 | 3.590 | 98.204 | 93.876 |
| llama3 | Ours | 5.419 | 96.902 | 88.443 | 3.136 | 97.591 | 90.933 | 4.035 | 98.117 | 93.601 |
| qwen2 | Ours | 7.828 | 96.179 | 85.977 | 3.857 | 97.431 | 90.697 | 5.236 | 97.973 | 93.510 |
| Ours w/o KF | Ours | 4.661 | 97.734 | 91.766 | 2.455 | 98.386 | 94.180 | 3.456 | 98.704 | 95.844 |

| Method | ASR Model | NarrativeQA | | | TAT-QA | | |
|---|---|---|---|---|---|---|---|
| | | WER↓ | SIM↑ | Pass↑ | WER↓ | SIM↑ | Pass↑ |
| - | canary | 8.880 | 92.731 | 71.408 | 18.608 | 76.620 | 21.285 |
| - | whisper | 8.527 | 92.913 | 72.425 | 19.981 | 77.032 | 21.781 |
| - | parakeet | 8.387 | 92.003 | 70.888 | 18.441 | 78.119 | 20.989 |
| - | Ours | 6.214 | 95.363 | 81.463 | 18.410 | 80.022 | 23.989 |
| phi3 | Ours | 4.255 | 97.283 | 89.105 | 7.740 | 94.520 | 76.124 |
| llama3 | Ours | 5.519 | 96.836 | 87.405 | 9.092 | 94.501 | 76.376 |
| qwen2 | Ours | 6.347 | 96.832 | 87.958 | 17.274 | 89.209 | 52.795 |
| Ours w/o KF | Ours | 4.320 | 97.891 | 91.970 | 6.510 | 96.652 | 85.949 |

| Rewrite LLM | ASR Model | SQUAD1.1 | | | SQUAD2.0 | | |
|---|---|---|---|---|---|---|---|
| | | WER↓ | SIM↑ | Pass↑ | WER↓ | SIM↑ | Pass↑ |
| - | canary | 10.186 | 89.682 | 64.276 | 9.553 | 90.373 | 64.787 |
| - | whisper | 9.672 | 90.488 | 65.755 | 9.711 | 90.537 | 65.790 |
| - | parakeet | 9.788 | 89.301 | 64.652 | 9.139 | 89.959 | 65.044 |
| - | Ours | 7.799 | 92.755 | 71.883 | 7.787 | 92.777 | 71.992 |
| phi3 | Ours | 5.509 | 95.983 | 84.439 | 5.473 | 95.982 | 84.305 |
| llama3 | Ours | 6.938 | 95.947 | 84.358 | 6.898 | 95.941 | 84.364 |
| qwen2 | Ours | 8.510 | 95.368 | 82.319 | 8.513 | 95.378 | 82.385 |
| Ours w/o KF | Ours | 5.972 | 97.094 | 88.972 | 5.937 | 97.083 | 88.862 |

Table 20: The quality evaluation results of our proposed method in datasets. The similarity here is the average of the results calculated by gte-large-en-v1.5 and stella_en_400M_v5 (Appendix B.4).

| Method | ASR Model | DROP | | | Quoref | | | ROPES | | |
|---|---|---|---|---|---|---|---|---|---|---|
| | | WER↓ | SIM↑ | Pass↑ | WER↓ | SIM↑ | Pass↑ | WER↓ | SIM↑ | Pass↑ |
| - | canary | 10.928 | 92.635 | 70.300 | 5.455 | 94.912 | 79.502 | 6.971 | 95.985 | 84.722 |
| - | whisper | 11.175 | 92.687 | 71.317 | 5.322 | 94.987 | 80.047 | 7.087 | 95.713 | 83.916 |
| - | parakeet | 10.416 | 92.066 | 70.108 | 5.084 | 94.452 | 79.984 | 6.208 | 95.482 | 85.655 |
| - | Ours | 9.191 | 94.344 | 76.890 | 4.141 | 96.511 | 85.858 | 5.203 | 97.304 | 90.525 |
| phi3 | Ours | 4.834 | 97.657 | 91.588 | 2.524 | 98.251 | 93.725 | 3.663 | 98.471 | 95.029 |
| llama3 | Ours | 5.318 | 97.707 | 92.249 | 3.118 | 98.070 | 93.243 | 4.110 | 98.412 | 95.103 |
| qwen2 | Ours | 7.510 | 96.827 | 89.049 | 3.726 | 97.781 | 92.370 | 5.202 | 98.209 | 94.590 |
| Ours w/o KF | Ours | 4.421 | 98.352 | 94.907 | 2.373 | 98.714 | 95.999 | 3.509 | 98.902 | 96.723 |

| Method | ASR Model | NarrativeQA | | | TAT-QA | | |
|---|---|---|---|---|---|---|---|
| | | WER↓ | SIM↑ | Pass↑ | WER↓ | SIM↑ | Pass↑ |
| - | canary | 8.880 | 94.134 | 76.648 | 18.608 | 80.533 | 23.363 |
| - | whisper | 8.527 | 94.165 | 77.020 | 19.981 | 80.827 | 24.241 |
| - | parakeet | 8.387 | 93.029 | 74.910 | 18.441 | 80.895 | 22.407 |
| - | Ours | 6.246 | 96.233 | 85.490 | 18.406 | 83.036 | 27.163 |
| phi3 | Ours | 4.255 | 97.891 | 92.533 | 7.607 | 95.700 | 83.149 |
| llama3 | Ours | 5.484 | 97.508 | 91.083 | 8.929 | 95.704 | 83.975 |
| qwen2 | Ours | 6.173 | 97.312 | 90.745 | 17.060 | 91.020 | 60.569 |
| Ours w/o KF | Ours | 4.198 | 98.352 | 94.730 | 6.241 | 97.362 | 90.644 |

| Rewrite LLM | ASR Model | SQUAD1.1 | | | SQUAD2.0 | | |
|---|---|---|---|---|---|---|---|
| | | WER↓ | SIM↑ | Pass↑ | WER↓ | SIM↑ | Pass↑ |
| - | canary | 10.186 | 91.212 | 67.328 | 9.553 | 91.919 | 67.956 |
| - | whisper | 9.672 | 91.886 | 68.469 | 9.711 | 91.932 | 68.413 |
| - | parakeet | 9.788 | 90.633 | 67.245 | 9.139 | 91.305 | 67.735 |
| - | Ours | 7.759 | 93.870 | 74.947 | 7.754 | 93.897 | 74.958 |
| phi3 | Ours | 5.460 | 96.701 | 87.152 | 5.404 | 96.705 | 87.093 |
| llama3 | Ours | 6.825 | 96.674 | 87.268 | 6.816 | 96.671 | 87.255 |
| qwen2 | Ours | 8.248 | 95.982 | 84.654 | 8.249 | 95.988 | 84.734 |
| Ours w/o KF | Ours | 5.802 | 97.624 | 91.259 | 5.791 | 97.610 | 91.219 |

Table 21: The quality evaluation results of our proposed method in datasets. The similarity here is the average of the results calculated by mxbai-embed-large-v1 and stella_en_400M_v5 (Appendix B.4).

| Method | ASR Model | DROP | | | Quoref | | | ROPES | | |
|---|---|---|---|---|---|---|---|---|---|---|
| | | WER↓ | SIM↑ | Pass↑ | WER↓ | SIM↑ | Pass↑ | WER↓ | SIM↑ | Pass↑ |
| - | canary | 10.928 | 92.104 | 68.601 | 5.455 | 94.977 | 79.902 | 6.971 | 95.734 | 84.282 |
| - | whisper | 11.175 | 92.148 | 69.580 | 5.322 | 95.035 | 80.384 | 7.087 | 95.381 | 83.733 |
| - | parakeet | 10.416 | 91.585 | 68.971 | 5.084 | 94.241 | 80.102 | 6.208 | 95.190 | 85.427 |
| - | Ours | 9.091 | 93.926 | 75.143 | 4.164 | 96.520 | 86.077 | 5.186 | 97.094 | 90.287 |
| phi3 | Ours | 4.876 | 97.313 | 90.420 | 2.526 | 98.209 | 93.707 | 3.689 | 98.343 | 94.901 |
| llama3 | Ours | 5.332 | 97.358 | 90.940 | 3.110 | 98.037 | 93.389 | 4.073 | 98.269 | 94.929 |
| qwen2 | Ours | 7.527 | 96.458 | 87.577 | 3.707 | 97.773 | 92.816 | 5.251 | 98.056 | 94.443 |
| Ours w/o KF | Ours | 4.479 | 98.056 | 93.859 | 2.352 | 98.673 | 95.980 | 3.483 | 98.805 | 96.732 |

| Method | ASR Model | NarrativeQA | | | TAT-QA | | |
|---|---|---|---|---|---|---|---|
| | | WER↓ | SIM↑ | Pass↑ | WER↓ | SIM↑ | Pass↑ |
| - | canary | 8.880 | 93.854 | 75.658 | 18.199 | 83.984 | 27.980 |
| - | whisper | 8.527 | 93.928 | 76.183 | 19.981 | 81.866 | 24.833 |
| - | parakeet | 8.387 | 93.056 | 74.910 | 18.441 | 82.205 | 23.554 |
| - | Ours | 6.213 | 96.071 | 84.898 | 18.608 | 81.803 | 24.094 |
| phi3 | Ours | 4.243 | 97.784 | 92.123 | 7.662 | 95.750 | 81.619 |
| llama3 | Ours | 5.394 | 97.381 | 90.575 | 8.962 | 95.706 | 82.141 |
| qwen2 | Ours | 6.084 | 97.200 | 90.335 | 16.919 | 90.915 | 59.847 |
| Ours w/o KF | Ours | 4.145 | 98.257 | 94.463 | 6.205 | 97.434 | 90.270 |

| Rewrite LLM | ASR Model | SQUAD1.1 | | | SQUAD2.0 | | |
|---|---|---|---|---|---|---|---|
| | | WER↓ | SIM↑ | Pass↑ | WER↓ | SIM↑ | Pass↑ |
| - | canary | 10.186 | 90.958 | 66.916 | 9.553 | 91.657 | 67.379 |
| - | whisper | 9.672 | 91.614 | 67.941 | 9.711 | 91.659 | 67.922 |
| - | parakeet | 9.788 | 90.346 | 66.793 | 9.139 | 91.007 | 67.177 |
| - | Ours | 7.723 | 93.661 | 74.226 | 7.726 | 93.682 | 74.242 |
| phi3 | Ours | 5.424 | 96.565 | 86.757 | 5.378 | 96.570 | 86.638 |
| llama3 | Ours | 6.765 | 96.505 | 86.738 | 6.766 | 96.501 | 86.706 |
| qwen2 | Ours | 8.155 | 95.810 | 84.055 | 8.158 | 95.817 | 84.126 |
| Ours w/o KF | Ours | 5.733 | 97.487 | 90.908 | 5.719 | 97.476 | 90.841 |

# D  MORE SPEAKER INFORMATION

As shown in Table 22, we provide more examples of speaker descriptions. These descriptions are presented to Parler in a variety of speech styles.

Table 22: Additional examples of speaker descriptions.

| Name | Gender | Position | Speech Rate | Clarity | Accent | Speaker Pitch | Description |
|---|---|---|---|---|---|---|---|
| Melvin | male | close-sounding | slowly | quite clean | English | very expressive and animated | A male voice with an English accent speaks slowly, with a close-sounding and quite clean delivery. The speaker's pitch is very expressive and animated, adding a vibrant and dynamic quality to the recording while maintaining good clarity. |
| Igor | male | close-sounding | slowly | quite clean | Pakistani | very expressive and animated | A male voice with a Pakistani accent speaks slowly, with a close-sounding and quite clean delivery. The speaker's pitch is very expressive and animated, adding a vibrant and dynamic quality to the recording while maintaining good clarity |
| Samuel | male | close-sounding | slowly | quite clean | Italian | very expressive and animated | A male voice with an Italian accent speaks slowly, with a close-sounding and quite clean delivery. The speaker's pitch is very expressive and animated, infusing the recording with a dynamic and lively quality while maintaining good clarity. |
| Joey | male | close-sounding | slowly | quite clean | Canadian | slightly expressive and animated | A male voice with a Canadian accent speaks slowly. The speaker's voice is close-sounding and quite clean, with a slightly expressive and animated pitch. The recording captures the speaker's subtle vocal nuances with good clarity. |
| Sherard | male | close-sounding | slowly | quite clean | Chinese | monotone | A male voice with a Chinese accent speaks slowly. The speaker's voice is close-sounding and quite clean, with a monotone pitch. The recording captures the speaker's steady, unvaried tone with clear definition. |
| Wyman | male | close-sounding | normally | quite clean | American | very expressive and animated | A male voice with an American accent speaks normally. The speaker's voice is close-sounding and quite clean, with a very expressive and animated pitch. The recording captures the speaker's dynamic vocal quality with clear, engaging detail. |
| Beatrix | female | close-sounding | slowly | quite clean | English | slightly expressive and animated | A female voice with an English accent speaks slowly. The speaker's voice is close-sounding and quite clean, with a slightly expressive and animated pitch. The recording captures the voice with a clear and engaging quality. |
| Jeanne | female | close-sounding | normally | quite clean | South African | slightly expressive and animated | A female voice with a South African accent speaks at a normal rate. The speaker's voice is close-sounding and quite clean, with a slightly expressive and animated pitch. The recording is clear and lively. |
| Amiable | female | close-sounding | quickly | quite clean | Pakistani | slightly expressive and animated | A female voice with a Pakistani accent speaks quickly. The speaker's voice is close-sounding and quite clean, with a slightly expressive and animated pitch. The recording is clear and lively. |
| Harmony | female | close-sounding | quickly | quite clean | Indian | slightly expressive and animated | A female voice with an Indian accent speaks quickly. The speaker's voice is close-sounding and quite clean, with a slightly expressive and animated pitch. The recording is clear and lively. |
| Alanna | female | close-sounding | slowly | quite clean | South African | very expressive and animated | A female voice with a South African accent speaks slowly. The speaker's voice is close-sounding and quite clean, with a very expressive and animated pitch. The recording captures the voice with a clear and vibrant quality. |
| Kirstyn | female | close-sounding | slowly | quite clean | Indian | very expressive and animated | A female voice with an Indian accent speaks slowly. The voice is close-sounding and quite clean, with a very expressive and animated pitch. The recording captures a dynamic and lively tone. |

## E  ASSETS AND LICENSES

Below, we provide the access links and open-source licenses for the models, datasets and main code used in this paper.

### E.1  MODELS

- Qwen2-7B-Instruct
    - Download URL:
      `https://huggingface.co/Qwen/Qwen2-7B-Instruct`
    - License: Apache-2.0
      `https://choosealicense.com/licenses/apache-2.0/`
- Phi-3-small-8k-instruct
    - Download URL:
      `https://huggingface.co/microsoft/Phi-3-small-8k-instruct`
    - License: MIT
      `https://choosealicense.com/licenses/mit/`
- Meta-Llama-3-8B-Instruct
    - Download URL:
      `https://huggingface.co/meta-llama/Meta-Llama-3-8B-Instruct`
    - License: llama3
      `https://llama.meta.com/llama3/license`
- parler-tts-large-v1
    - Download URL:
      `https://huggingface.co/parler-tts/parler-tts-large-v1`
    - License: Apache-2.0
      `https://choosealicense.com/licenses/apache-2.0/`
- whisper-large-v3
    - Download URL:
      `https://huggingface.co/openai/whisper-large-v3`
    - License: Apache-2.0
      `https://choosealicense.com/licenses/apache-2.0/`
- canary-1b
    - Download URL:
      `https://huggingface.co/nvidia/canary-1b`
    - License: CC-BY-NC-4.0
      `https://spdx.org/licenses/CC-BY-NC-4.0`
- parakeet-tdt-1.1b
    - Download URL:
      `https://huggingface.co/nvidia/parakeet-tdt-1.1b`
    - License: CC-BY-4.0
      `https://choosealicense.com/licenses/cc-by-4.0/`
- gte-large-en-v1.5
    - Download URL:
      `https://huggingface.co/Alibaba-NLP/gte-large-en-v1.5`
    - License: Apache-2.0
      `https://choosealicense.com/licenses/apache-2.0/`
- mxbai-embed-large-v1
    - Download URL:
      `https://huggingface.co/mixedbread-ai/mxbai-embed-large-v1`

– License: Apache-2.0
  `https://choosealicense.com/licenses/apache-2.0/`
- stella_en_400M_v5
  – Download URL:
    `https://huggingface.co/dunzhang/stella_en_400M_v5`
  – License: MIT
    `https://choosealicense.com/licenses/mit/`

## E.2 DATASETS

- TAT-QA
  – Download URL:
    `https://github.com/NExTplusplus/TAT-QA/tree/master/dataset_raw`
  – License: CC-BY-4.0
    `https://creativecommons.org/licenses/by/4.0/`
- DROP
  – Download URL:
    `https://huggingface.co/datasets/ucinlp/DROP`
  – License: CC-BY-SA-4.0
    `https://choosealicense.com/licenses/cc-by-sa-4.0/`
- SQUAD1.1
  – Download URL:
    `https://huggingface.co/datasets/rajpurkar/squad`
  – License: CC-BY-SA-4.0
    `https://choosealicense.com/licenses/cc-by-sa-4.0/`
- SQUAD2.0
  – Download URL:
    `https://rajpurkar.github.io/SQuAD-explorer/`
  – License: CC-BY-SA-4.0
    `https://choosealicense.com/licenses/cc-by-sa-4.0/`
- ROPES
  – Download URL:
    `https://huggingface.co/datasets/allenai/ropes`
  – License: CC-BY-SA-4.0
    `https://choosealicense.com/licenses/cc-by-4.0/`
- NarrativeQA
  – Download URL:
    `https://huggingface.co/datasets/deepmind/narrativeqa`
  – License: Apache-2.0
    `https://choosealicense.com/licenses/apache-2.0/`
- Quoref
  – Download URL:
    `https://huggingface.co/datasets/allenai/quoref`
  – License: CC-BY-4.0
    `https://creativecommons.org/licenses/by/4.0/`

## E.3 CODE

- VLLM
  – Download URL:
    `https://github.com/vllm-project/vllm`

- License: Apache-2.0
  `https://choosealicense.com/licenses/apache-2.0/`
- Transformers
  - Download URL:
    `https://github.com/huggingface/transformers`
  - License: Apache-2.0
    `https://choosealicense.com/licenses/apache-2.0/`
- pyTorch
  - Download URL:
    `https://github.com/pytorch/pytorch`
  - License: PyTorch
    `https://github.com/pytorch/pytorch?tab=`
    `License-1-ov-file#readme`

