# OpenReview forum: "Unlocking Speech Instruction Data Potential with Query Rewriting"
_ICLR.cc/2025/Conference — Submitted to ICLR 2025_

### Official Review · Reviewer_PEw1 · 2024-10-30

**Soundness:** 2
**Presentation:** 2
**Contribution:** 2
**Rating:** 3
**Confidence:** 4

**Summary:**

This paper addresses text normalization in Text-to-Speech (TTS) through Large Language Model (LLM) utilization, aiming to better capture contextual information. By leveraging multi-LLM query rewriting and multi-agent annotation, the authors propose an innovative data generation framework that enables end-to-end speech language models (LSLMs) to better follow spoken instructions. This approach aligns synthetic speech and textual information with TTS models, achieving improvements in data usability and quality while minimizing reliance on human annotation. However, the method remains primarily a data augmentation strategy limited to specific TTS model, and lacking comparision with typical text normalization method for TTS.

**Strengths:**

The paper tackles text normalization for TTS using a text-based LLM framework, potentially improving contextual understanding and data synthesis efficiency for LSLM training.

**Weaknesses:**

- Limited Scope: The focus is primarily on input text rewriting for TTS, which is essential for text normalization but less relevant to speech language models (SLMs) training. The framework generates instruction data but does not advance model training methods.

- Limited Technical Contribution: This work is mainly a data augmentation strategy, using TTS for producing instruction data in a speech-question/text-response format without introducing new training algorithms for LSLMs.

- Lack of Baseline Comparison: Text normalization in TTS has a rich history, especially in industrial applications. Prior studies, such as [1] and [2], propose alternative normalization techniques. The only rule-based baseline provided here involves basic numeric conversion, which oversimplifies comparison and limits validation of this method against established techniques.

- Insufficient Details on Model Training and Evaluation: The results section ("Use synthetic data to fine-tune LSLMs") lacks clarity regarding fine-tuning methods. For example, Table 4 lists "Qwen2-Audio-7B-Instruct" as the model, but it remains unclear if it was continually fine-tuned. Additionally, input-output specifics for this model are not provided.

- Dependency on Specific TTS Model: The study employs only Parler TTS for text synthesis. It’s uncertain how the framework would perform if a phoneme-based TTS model were used, particularly regarding out-of-vocabulary issues.

- ASR Model Dependency: The method’s efficacy in cases where ASR systems fail to recognize rare words is uncertain, raising questions about the proposed solution's handling of out-of-vocabulary cases


References
1. Ebden, Peter, and Richard Sproat. "The Kestrel TTS text normalization system." Natural Language Engineering, 21.3 (2015): 333-353.
2. Zhang, Hao, et al. "Neural models of text normalization for speech applications." Computational Linguistics, 45.2 (2019): 293-337.

**Questions:**

Some suggestions are listed below:

- Figure 1: Highlight differences using color or bold font for better visibility.

- Figure 2: Text readability is hindered by font size; should consider increasing.

- Table 8: The caption appears incorrect and should be reviewed for accuracy.

---

> ### Author Response · Authors · 2024-11-25
> **Response by Authors to Reviewer PEw1 (Part.1)**
>
> We are deeply grateful for your recognition of our work's innovation and thoroughness, as well as their constructive feedback. We have addressed each suggestion on the manuscript's weaknesses and made the necessary revisions.
>
> > Q1: Limited Scope:...
> >
> > Q2: Limited Technical Contribution:...
>
>
> The query rewriting framework proposed in this paper differs from text normalization. Text normalization does not alter the word order or expression of the query itself, whereas query rewriting can rephrase sentences in new ways. This capability allows query rewriting to overcome the limitations of text normalization methods when dealing with complex texts or texts requiring additional knowledge.
>
>
> The primary contribution of this paper in terms of training methodology lies in the discussion of training objectives. Previous works have used the continuation results generated by LLMs from the textual information of automatic speech recognition data as training objectives.
> In contrast, this paper highlights the significant differences in results when using high-quality answers versus LLM-generated continuations as training objectives, demonstrating the limitations of existing continuation-based methods.
>
>
> > Q3: Insufficient Details on Model Training and Evaluation: The results section ("Use synthetic data to fine-tune LSLMs") lacks clarity regarding fine-tuning methods...
>
> We will supplement the following details in the experimental setup section:
>
> ```text
> For the training in LSLM Finetune, we set r=8 and a=32. The peak learning rate was set to 3e-5, and the cosine scheduler was used for learning rate adjustment.
> ```
>
> The instruction templates we used during the training phase are as follows. We will include this information in the appendix of the revised manuscript.
>
> ```text
> SYSTEM
>
> This is a chat between a user and an artificial intelligence assistant. The assistant gives helpful, detailed, and polite answers to the user’s questions based on the context. The assistant should also indicate when the answer cannot be found in the context.
>
> {doc}
>
> User
>
> Answer the following question with a short span. The answer needs to be just in a few words.\n
> ```
>
>
> > Q4: ASR Model Dependency: The method’s efficacy in cases where ASR systems fail to recognize rare words is uncertain, raising questions about the proposed solution's handling of out-of-vocabulary cases
>
> For the out-of-vocabulary issues, through the query rewriting method, we can address out-of-vocabulary terms such as abbreviations, dates, and mathematical formulas.

---

> ### Author Response · Authors · 2024-11-25
> **Response by Authors to Reviewer PEw1 (Part.2)**
>
> > Q5: Lack of Baseline Comparison: Text normalization in TTS has a rich history...
> >
> > Q6: Dependency on Specific TTS Model: The study employs only Parler TTS for text synthesis. It’s uncertain how the framework would perform if a phoneme-based TTS model were used, particularly regarding out-of-vocabulary issues.
>
> We added a text normalization method as a baseline for comparison, which achieved 98% accuracy in the Google Text Normalization Challenge.
>
> To demonstrate the generality of our method and control the influence of speaker style descriptions on the synthesis results, the TTS models in the following experiments use a single speaker style setting during synthesis.
>
> The experimental results are shown below, where we use SIM as a metric to evaluate the results. The experimental results demonstrate that our method consistently achieves favorable performance across different TTS models.
>
> |   TTS   Model     	   | Method   	          |  Drop    	  | narrativeqa 	 | Quoref   	  |  ropes   	  | squad1.1  	 | squad2.0  	 |  tatqa   	  | Average  	  |
> |:---------------------:|---------------------|:-----------:|:-------------:|:-----------:|:-----------:|:-----------:|:-----------:|:-----------:|:-----------:|
> |    MeloTTS       	    | Original 	          | 96.61    	  |  95.13    	   | 97.27    	  | 98.22    	  | 95.75    	  | 93.48    	  | 97.43    	  | 96.27    	  |
> |           	           | TN       	          | 97.50    	  |  96.36    	   | 98.00    	  | 98.83    	  | 94.86    	  | 95.54    	  | 97.96    	  | 97.01    	  |
> |           	           | Phi3     	          | 97.57    	  |  95.82    	   | 97.81    	  | 98.70    	  | 96.74    	  | 95.71    	  | 98.29    	  | 97.23    	  |
> |           	           | Qwen2    	          | 97.59    	  |  96.42    	   | 98.01    	  | 98.85    	  | 96.98    	  | 95.68    	  | 98.19    	  | 97.39    	  |
> |           	           | Llama3   	          | 97.65    	  |  96.32    	   | 98.01    	  | 98.91    	  | 97.10    	  | 95.78    	  | 98.30    	  | 97.44    	  |
> |           	           | Ours  w/o KF   	    | **98.19** 	 |  **97.02** 	  | **98.37** 	 | **99.12** 	 | **97.59** 	 | **96.80** 	 | **98.53** 	 | **97.94** 	 |
> |   mms-tts-eng     	   | Original 	          | 92.05    	  |  91.46    	   | 93.77    	  | 93.42    	  | 91.28    	  | 91.23    	  | 84.61    	  | 91.12    	  |
> |           	           | TN       	          | 94.22    	  |  **93.82** 	  | 95.64    	  | 95.14    	  | 93.86    	  | 93.26    	  | 90.75    	  | 93.81    	  |
> |           	           | Phi3     	          | 95.06    	  |  93.00    	   | 95.46    	  | 94.97    	  | 93.90    	  | 93.86    	  | 92.52    	  | 94.11    	  |
> |           	           | Qwen2    	          | 95.18    	  |  91.47    	   | 95.51    	  | 95.04    	  | 94.32    	  | 94.62    	  | 91.81    	  | 93.99    	  |
> |           	           | Llama3   	          | 94.52    	  |  91.47    	   | 95.51    	  | 94.98    	  | 94.85    	  | 94.82    	  | 94.30    	  | 94.35    	  |
> |           	           | Ours  w/o KF      	 | **96.86** 	 |  93.01    	   | **96.51** 	 | **95.95** 	 | **95.96** 	 | **96.07** 	 | **96.17** 	 | **95.79** 	 |
> | Parler-TTS-Large-v1 	 | Original 	          | 93.04    	  |  95.26    	   | 95.51    	  | 96.01    	  | 92.75    	  | 91.99    	  | 81.63    	  | 92.31    	  |
> |           	           | TN       	          | 96.41    	  |  97.39    	   | 97.63    	  | 98.06    	  | 95.89    	  | 95.61    	  | 89.07    	  | 95.72    	  |
> |           	           | Phi3     	          | 97.21    	  |  97.40    	   | 97.95    	  | 97.97    	  | 96.49    	  | 96.19    	  | 93.80    	  | 96.72    	  |
> |           	           | Qwen2    	          | 97.17    	  |  97.51    	   | 97.93    	  | 98.05    	  | 96.29    	  | 95.64    	  | 91.63    	  | 96.32    	  |
> |           	           | Llama3   	          | 97.41    	  |  97.41    	   | 97.78    	  | 96.74    	  | 96.74    	  | 96.44    	  | 96.01    	  | 96.93    	  |
> |           	           | Ours  w/o KF     	  | **98.09** 	 |  **98.19** 	  | **98.52** 	 | **98.30** 	 | **97.31** 	 | **97.32** 	 | **97.21** 	 | **97.85** 	 |
>
> Which *TN* mean the text normalization, *Phi3* mean only use Phi-3-small-8k-instruct to rewrite query, *Qwen2* mean only use Qwen2-7B-Instruct to rewrite query, *Llama3* mean only use Llama-3-8B-Instructt to rewrite query,
> and *Ours  w/o KF* refers to using only the training-free rewriting framework illustrated in Figure 1.
>
> For the out-of-vocabulary issues, through the query rewriting method, we can address out-of-vocabulary terms such as abbreviations, dates, and mathematical formulas.

---

> ### Author Response · Authors · 2024-11-29
> **Summary of response and look forward to the feedback**
>
> We greatly appreciate the thoughtful critique and suggestions from Reviewer PEw1. Below is a summary of our revisions and clarifications based on the provided feedback.
>
> 1.Technical Contributions. This paper contributes to both data synthesis methods and training approaches. We propose a novel synthesis method for speech command datasets, which, unlike text normalization, enhances the quality of synthesized speech commands by rewriting queries while maintaining semantic consistency.
> In terms of training methods, our comparison of different training strategies on downstream tasks highlights that using high-quality human-annotated answers as training targets is crucial for improving the performance of LSLM. Methods based on LLM continuation limit the model’s ability to follow speech commands effectively.
>
> 2.Baseline Comparison. The work most closely related to ours is text normalization, a common industrial approach used to optimize the input for TTS models, which rewriting queries to fit the input requirements of the TTS model without altering their expression.
> We added a text normalization method as a baseline for comparison, which achieved 98% accuracy in the Google Text Normalization Challenge. We present the experimental results in Tables 2, 3, 14, and 15 of the revised manuscript.
>
> 3.Scalability. We considered both multi-speaker and single-speaker settings, adding two TTS models: MeloTTS and MMS-TTS-ENG.
> The experimental results under the multi-speaker setting are presented in Tables 2 and 3, while the results for the single-speaker setting are shown in Tables 14 and 15.
> The results validate the excellent scalability of our method and demonstrate its unique advantages, bridging the gap in synthesis accuracy between vocoder-based TTS models and autoregressive TTS models.
>
> 4.Experimental Details. In the revised manuscript, we provide detailed experimental setups in Section 5.1, including baselines, datasets, training methods, backbone models, and training details. Additionally, we present the instruction templates used during the fine-tuning phase of LSLM in Table 11.
>
>  We hope these revisions and clarifications address your concerns and look forward to any additional feedback or questions.

---

> > ### Author Response · Authors · 2024-11-30
> > **We are keen to discuss further with you**
> >
> > Dear Reviewer PEw1,
> >
> > Thank you for your valuable time and the constructive feedback you have provided. We genuinely hope reviewer tPgc could kindly check our response.
> >
> > As the deadline for the discussion period is nearing, we would greatly appreciate it if you could kindly let us know whether there are any further questions. Thank you!
> >
> > Best wishes,
> >
> > Authors

---

### Official Review · Reviewer_FVbN · 2024-11-02

**Soundness:** 3
**Presentation:** 2
**Contribution:** 2
**Rating:** 5
**Confidence:** 4

**Summary:**

The paper addresses the enhancement of instruction-following capabilities in speech language models through the creation of an improved synthetic speech instruction dataset. To achieve this, the authors propose a method to rewrite text instructions, aligning them more closely with the typical training distribution of TTS models. This rewriting framework utilizes multiple LLMs and incorporates annotation and validation of synthesized speech via a combination of ASR and embedding methods. Experimental results on several text-based QA datasets demonstrate that this approach enhances the usability of synthetic data and improves the preservation of linguistic information within synthesized speech.

**Strengths:**

1. Generating high-quality synthetic speech instruction data is a valuable task for the community, yet it remains relatively underexplored.
2. The paper introduces a versatile framework that integrates existing LLMs, ASR, and text embedding methods in a plug-and-play manner, requiring no additional training aside from the knowledge fusion component.
3. Results from automatic metrics indicate that the proposed framework enhances the quality and usability of synthetic data.

**Weaknesses:**

Quality

1. The paper begins by highlighting the gap between LLM-generated responses and human responses, but it’s unclear if their framework effectively addresses this. Human responses often include disfluencies or may involve mid-sentence question reformulations. A more rigorous human-in-the-loop evaluation would be beneficial to assess if fine-tuning on their synthetic speech data genuinely enhances the speech language model's ability to follow "spoken" human instructions.
2. The quality of synthetic data is primarily assessed using automated metrics. While I recognize the potential cost, a relevance judgment by human evaluators could provide valuable insights into whether the synthetic data aligns with how humans would expect to interact with a speech model. Given that the core objective is to improve the model's ability to follow human instructions, human-in-the-loop evaluation is essential.

Clarity

1. The paper could benefit from improved readability, particularly with the equations. Including backpointers to the sections where variables are first introduced would aid in following the mathematical formulation.
2. There are typos, such as "finitune" on page 8. A thorough proofreading is recommended to address these grammatical issues.
3. It’s unclear whether all embedding methods in Table 5 are used, or if only a subset based on Table 6 is applied.
4. The reference to "71% to 93%" lacks context. Clarifying what this range represents would improve comprehension.

**Questions:**

Refer to weaknesses

---

> ### Author Response · Authors · 2024-11-25
> **Response by Authors to Reviewer FVbN (Part.1)**
>
> We are deeply grateful for your recognition of our work's innovation and thoroughness, as well as their constructive feedback. We have addressed each suggestion on the manuscript's weaknesses and made the necessary revisions.
>
> > Q1: The paper begins by highlighting the gap between LLM-generated responses and human responses...
>
> The response gap between the LLM and humans mentioned in this paper primarily refers to the differences between text responses generated by the LLM and those annotated by humans.
> Our experiments in Table 4 demonstrate that using LLM-generated responses as training targets is insufficient compared to using high-quality, human-annotated responses.
>
>
>
>
> > Q2: The quality of synthetic data is primarily assessed using automated metrics...
>
> Thank you for your suggestion.
> To more thoroughly evaluate the semantic consistency between our rewritten content and the original, we employed a voting mechanism using multiple LLMs to simulate human annotation.
> This approach verifies the consistency of linguistic information between the synthesized speech generated from the rewritten queries and the original text.
>
> Specifically, we used Qwen2-7B-Instruct, Llama-3-8B-Instruct, and Phi-3-small-8k-instruct to validate consistency, employing the following prompts to guide the models in providing judgment results.
> We conducted tests on the Drop, Quoref, ropes, and TAT-QA datasets, utilizing Parler-TTS-Large-v1 as TTS Model.
>
>
> ```text
> # Task #
>
> Please evaluate whether the following two sentences convey identical meanings in content. Use yes or no to give your verdict.
>
> # Sentence 1 #
>
> {original_text}
>
> # Sentence 2 #
>
> {rewrite_text}
> ```
> *original_text* refers to the original text, while *rewrite_text* refers to the linguistic information of the speech synthesized via TTS after rewriting, which is obtained through an ASR model.
>
>
> The following are the experimental results. We use the pass rate as the metric, where a sentence is considered to pass the validation if more than two models deem it consistent.
>
>
> | Method   	 | Drop  	  | tatqa 	  | Quoref 	 | ropes 	  | Average 	 |
> |------------|:--------:|:--------:|:--------:|:--------:|:---------:|
> | Original 	 | 71.20  	 | 22.82  	 | 83.84  	 | 78.28  	 | 64.03  	  |
> | TN       	 | 85.37  	 | 41.41  	 | 92.09  	 | 87.75  	 | 76.66  	  |
> | Phi3     	 | 92.98  	 | 77.58  	 | 95.31  	 | 89.27  	 | 88.79  	  |
> | Qwen2    	 | 93.74  	 | 77.45  	 | 95.60  	 | 90.49  	 | 89.32  	  |
> | Llama3   	 | 94.49  	 | 79.61  	 | 93.96  	 | 87.38  	 | 88.86  	  |
> | Ours     	 | 97.42  	 | 97.41  	 | 97.89  	 | 94.02  	 | 96.69  	  |
>
>
> > Q3: It’s unclear whether all embedding methods in Table 5 are used, or if only a subset based on Table 6 is applied.
>
> In Table 6, we follow the method illustrated in Figure 1, using the mean of all embedding models as the similarity evaluation metric.
> We report the results of using different combinations of embedding models in Appendix C to verify that our method does not depend on specific embedding models.
>
> > Q4: The reference to "71% to 93%" lacks context. Clarifying what this range represents would improve comprehension.
>
> The usability calculation results are derived from the mean *PASS* scores in the experiments, which are distributed across three tables in Tabel 5.
> To clarify the source of these results, we have consolidated this metric into the table below. We will include the following table in the appendix of the revised PDF.
>
>
> |  Method   	   | DROP  	  | Quoref 	 | ROPES 	  | NarrativeQA 	 | TAT-QA 	 | SQUAD1.1 	 | SQUAD2.0 	 | Average 	 |
> |:-------------:|:--------:|:--------:|:--------:|:-------------:|:--------:|:----------:|:----------:|:---------:|
> |  Original  	  | 74.40  	 | 84.78  	 | 89.80  	 |  83.63    	   | 25.85  	 | 73.39   	  | 73.45   	  | 72.19   	 |
> |   Phi3    	   | 89.82  	 | 92.69  	 | 94.60  	 |  91.04    	   | 79.46  	 | 73.39   	  | 73.45   	  | 84.92   	 |
> |  Qwen2    	   | 87.28  	 | 91.70  	 | 94.09  	 |  89.38    	   | 56.63  	 | 83.46   	  | 83.53   	  | 83.72   	 |
> |  Llama3   	   | 90.40  	 | 92.23  	 | 94.44  	 |  89.37    	   | 80.21  	 | 85.91   	  | 85.96   	  | 88.36   	 |
> | Ours w/o KF 	 | 93.43  	 | 95.24  	 | 96.32  	 |  93.54    	   | 88.72  	 | 90.19   	  | 90.19   	  | 92.52   	 |
> |   Ours    	   | 94.11  	 | 95.59  	 | 96.71  	 |  94.29    	   | 89.12  	 | 90.78   	  | 90.93   	  | 93.07   	 |

---

> > ### Comment · Reviewer_FVbN · 2024-11-26
> >
> > Thanks for your response. Since my primary concern about a more rigorous human-in-the-loop evaluation to evaluate the quality of synthetic data has not been addressed, I'll keep my current rating.

---

> > > ### Author Response · Authors · 2024-11-27
> > > **Response by Authors to Reviewer FVbN (Part.2)**
> > >
> > > Thank you very much for your response. We value your insights, and in response to your points of concern, we offer the following explanations:
> > >
> > > As shown in the overhead calculations provided in our paper, using a rigorous human-machine interaction approach to evaluate the quality of the entire dataset fully is extremely challenging, implying high experimental costs and lengthy annotation times.
> > >
> > > We conducted a small-scale manual annotation experiment to demonstrate the improvement in data quality achieved by our method under rigorous human-machine interaction evaluation.
> > >
> > > Specifically, we randomly selected 100 samples from each of the four datasets: DROP, Quoref, ROPES, and NarrativeQA.  We ensured that these samples had a SIM score below 50 under the original method, which indicates that there are significant differences in linguistic information between the speech data and the original text.
> > >
> > > During the annotation process, we instructed annotators to verify whether the speech content conveyed the same meaning as the original text and provide annotation results.  Any annotations completed in less time than the duration of the audio were discarded and re-annotated to ensure the validity of the annotations.
> > >
> > > The following are our experimental results, demonstrating that our method significantly improves the linguistic consistency of synthesized speech.
> > >
> > > | Method  	  | DROP 	 | Quoref 	 | ROPES 	 | NarrativeQA 	 |
> > > |:----------:|:------:|:--------:|:-------:|:-------------:|
> > > | Original 	 |  0  	  |  0   	   |  0   	  |   0      	    |
> > > |  TN    	   | 34  	  |  47   	  |  52  	  |   49     	    |
> > > |  Ours   	  | 74  	  |  79   	  |  69  	  |   71     	    |

---

> > > > ### Comment · Reviewer_FVbN · 2024-11-28
> > > >
> > > > 1. **Evaluation of Synthetic Data for Training Speech LMs**
> > > > I am primarily interested in understanding how your generated speech instruction data improves the efficacy of training speech LMs. Upon reviewing Table 4, I realize it addresses my interest in this aspect. However, I have a specific suggestion: in Table 4, could you also evaluate the generation quality of the speech LM trained on your synthetic data? Specifically, I would like to see how effectively it bridges the gap compared to using human-annotated speech continuation prompts.
> > > >
> > > > 2. **Writing Quality and Missing Experimental Details**
> > > > After carefully reviewing the paper, I found it still lacks clarity and contains numerous typos, which makes it challenging to follow. Additionally, there are missing details about the experimental setup. For example, in the results presented in Table 4, do you train separate speech LMs for each dataset? This crucial detail is unclear. Furthermore, please provide explicit metrics in the table captions. For instance, the caption of Table 4 does not clarify whether it evaluates the quality of the synthetic data or the generation quality of the speech LM.
> > > >
> > > > 3. **Human-in-the-Loop Evaluation**
> > > > Thank you for providing the additional results. I am particularly interested in a human-in-the-loop evaluation of the quality of the speech LMs trained on synthetic data. Such evaluation would align with prior work on generating synthetic speech instruction data (e.g., https://arxiv.org/pdf/2311.06753).
> > > >
> > > > 4. **Clarifications on Table 5 and Golden Targets**
> > > > Could you provide more details about the experimental setup in Table 5? Specifically:
> > > >
> > > > a. How were the golden targets generated? Generating human-annotated answers for 87K samples appears highly resource-intensive.
> > > >
> > > > b. How many annotators were involved in the process?
> > > >
> > > > c. Were multiple annotations collected for each question?

---

> > > > > ### Author Response · Authors · 2024-11-29
> > > > > **Response by Authors to Reviewer FVbN (Part.3)**
> > > > >
> > > > > > Evaluation of Synthetic Data for Training.....
> > > > >
> > > > > In Table 4, we present the performance on downstream tasks after training the model with data synthesized using the methods proposed in this paper.
> > > > >
> > > > > We provided the specific metrics in Section 5.2 and updated the title of Table 4 in the revised manuscript for better readability.
> > > > >
> > > > > > Writing Quality and Missing Experimental Details After carefully reviewing the paper, I found it still lacks clarity and contains numerous typos, which makes it challenging to follow. Additionally, there are missing details about the experimental setup. For example, in the results presented in Table 4, do you train separate speech LMs for each dataset? This crucial detail is unclear. Furthermore, please provide explicit metrics in the table captions. For instance, the caption of Table 4 does not clarify whether it evaluates the quality of the synthetic data or the generation quality of the speech LM.
> > > > >
> > > > > We train LSLMs using data synthesized under the MultiSpeaker Setting, combining all datasets to train it.
> > > > >
> > > > > We have added the data organization method for training LSLMs in lines 290–291 of the revised manuscript and provided an introduction to the training method in Appendix B.2.
> > > > >
> > > > > > Human-in-the-Loop Evaluation Thank you for providing the additional results. I am particularly interested in a human-in-the-loop evaluation of the quality of the speech LMs trained on synthetic data. Such evaluation would align with prior work on generating synthetic speech instruction data (e.g., https://arxiv.org/pdf/2311.06753).
> > > > >
> > > > > In Table 4, we present the performance on downstream tasks after training the model with data synthesized using the methods proposed in this paper. Since the tasks used in this study have a single correct answer, using ROUGE-L as the evaluation metric better reflects the model’s performance.
> > > > >
> > > > > To address your concerns regarding the human evaluation of generation quality, we conducted the following experiment:
> > > > >
> > > > > We selected 50 samples from Drop, Quoref, Ropes, and NarrativeQA, respectively, and had human annotators evaluate the quality of the responses. We ensured that the input speech did not contain any quality issues in the text.
> > > > >
> > > > > During the evaluation process, we asked annotators to assess whether the model-generated results were consistent with the reference answers. The experimental results are shown in the table below. We reported the number of samples that passed the verification.
> > > > >
> > > > >
> > > > >
> > > > > | Method         	 | Drop 	 | Quoref 	 | Ropes 	 | NarrativeQA 	 |
> > > > > |------------------|--------|----------|---------|---------------|
> > > > > | Continue+Ours  	 | 8    	 | 32     	 | 24    	 | 19          	 |
> > > > > | Golden+Oringai 	 | 7    	 | 29     	 | 25    	 | 21          	 |
> > > > > | Golden+Ours    	 | 20   	 | 41     	 | 29    	 | 31          	 |
> > > > >
> > > > >
> > > > > The method settings are as follows:
> > > > >
> > > > > * Continue+Ours: The training objective uses the Continue setting, and the data synthesis method uses our proposed approach.
> > > > > * Golden+Original: The training objective uses the Golden setting, and the data synthesis method uses the original setting.
> > > > > * Golden+Ours: The training objective uses the Golden setting, and the data synthesis method uses our proposed approach.
> > > > >
> > > > > These settings correspond to the training methods in rows 2–4 of Table 4.
> > > > >
> > > > >
> > > > > > Clarifications on Table 5 and Golden Targets Could you provide more details about the experimental setup in Table 5? Specifically:
> > > > >
> > > > > The seven datasets used in this paper all have high-quality answers officially provided, which we reused as responses.
> > > > >
> > > > > We have added more information about the Golden setting in lines 312–313 of the revised manuscript to reduce confusion. The specific updates are as follows:
> > > > >
> > > > > (1) Golden: Using high-quality human-annotated answers. The dataset used in this paper includes high-quality official annotations for responses, which we have reused;

---

> > > > > > ### Comment · Reviewer_FVbN · 2024-12-02
> > > > > >
> > > > > > Thank you for providing a detailed response and conducting multiple experiments. Based on your feedback, I have updated my soundness score and overall evaluation. The work presents promising results in aligning LLM outputs with TTS distributions. However, I have two suggestions for future work that could enhance the overall quality and impact of this research:
> > > > > >
> > > > > > 1. Incorporating Paralinguistic Features: To effectively bridge the gap between LLM-generated responses and human responses, it is crucial to include paralinguistic information in the audio. This could involve adding disfluencies, mid-sentence question reformulations, and other natural speech characteristics. Such additions would ensure that SpeechLM, trained on this synthetic data, is better equipped to handle these conditions during inference.
> > > > > >
> > > > > > 2. Improved Human-in-the-Loop Evaluation: I suggest the authors design a more comprehensive human-in-the-loop evaluation framework. In this setup, users could ask open-ended questions to SpeechLM and assess its ability to handle these queries while following user instructions. This approach would significantly strengthen the credibility and practical applicability of the work.
> > > > > >
> > > > > > Lastly, I recommend the authors proofread the paper thoroughly, as it still contains some typos. Additionally, more detailed and descriptive table captions would improve the paper’s readability.

---

> > > > > > > ### Author Response · Authors · 2024-12-04
> > > > > > >
> > > > > > > Dear Reviewer FVbN,
> > > > > > >
> > > > > > > Thank you for your positive response and support for our work! We noticed that the rating has not been updated, so we would like to confirm this with you.
> > > > > > >
> > > > > > > Thank you again for your time and assistance!

---

### Official Review · Reviewer_2kw3 · 2024-11-04

**Soundness:** 3
**Presentation:** 3
**Contribution:** 2
**Rating:** 5
**Confidence:** 4

**Summary:**

This paper presents a query rewriting framework for rewriting text queries such that TTS models are able to easily process and synthesize. To accomplish this, several LLMs are prompted to rewrite the input query such that things like abbreviations and formulas would be written in a way that a TTS model can understand. Multiple ASR models are used to filter the TTS outputs and match them against the original input to make sure that they are aligned. In order to increase the diversity of the speakers, an LLM is prompted to generate speaker descriptions which are passed to the TTS model to control the speaker identity and style. Knowledge Fusion is used to improve the model's performance on rewriting challenging queries. Experiments show that their technique improves the linguistic alignment between the text and generated speech while

**Strengths:**

Paper presents a detailed infrastructure for generation and filtering synthetic datasets for speech instruction datasets. Results show consistent improvement over naive TTS generation. Experimental results show across multiple datasets that the generated speech is higher quality across WER, SIM and PASS and that their technique improves downstream performance on NarrativeQA.

**Weaknesses:**

One of the main issues with this paper is the Main Result in Table 4 and Section 5.3. Authors only show that performance improves for exactly 1 model and on exactly 1 dataset. This is not sufficient evidence to claim that this technique extrapolates to other downstream tasks. This is very strange, since the authors clearly had all of these other datasets available that they could have used for evaluation.
This table is also not very well explained; there is no experimental setup section about how this experiment was performed.

Parts of this manuscript are poorly written and explained. Lots of typos and grammatical errors. (See questions)
In the second part of Section 5.2, the authors say they use ROUGE-L to "evaluate the quality of the generated results", however, it doesn't seem like this metric is used anywhere.

The implementation details in section 5.1 on page 7 also state that top-p for generation is 10 (I am assuming 10%), which is extremely low, this might as well be greedy sampling.

Furthermore the use of Knowledge Fusion is not very well explained. Section 4.3 states that they use Meta-Llama-3-8B-Instruct on line 266/267 as the backbone model to train "the model"? In Figure 3, it seems like they just train a Llama model, how does this integrate with the existing rewriting models in the results sections, where something like "Phi3+Llama3+Qwen2+KF" is written? How do these LLMs integrate with each other in the pipeline?

Furthermore, the abstract, introduction and conclusion all repeat this statement of improving data usability from "71% to 93%" (the conclusion actually says 91%)  but it is unclear where this number actually comes from.

Ablations are incomplete. In particular, it is unclear whether this technique improves over just having the LLMs rewrite the queries and using raw TTS after that.

**Questions:**

On line 053, you say that these training methods "treat the speech modality as files rather than instructions", what does this mean?
Furthermore, you state that the integration of "traditional tasks in the speech domain (such as speaker classification, speech entity recognition, etc.)" led to the trained models to "lack the ability to follow speech instructions and lead to hallucinations due to the severe bias in task distribution." Where is any reference for this phenomenon?

In Figure 1, it is stated that these are depictions of several recognition errors in ASR models. However, the second error: "Sentence Pattern Error" is strange, it doesn't seem like there is any error other than capitalization? Is the error that the bot didn't put the "?" at the end? But semantically, the text is a question, not a sentence, so I am not sure what the error here is.

Typos and grammatical mistakes throughout the manuscript:
* Extra period in Equation (1) on line 147, it should not be there since line 149 is the continuation of the sentence afterwards. Also an extra period in Equation (2), (3), (4)
* Line 161, "oringal" instead of "original"
* Incorrectly capitalized "We" near the end of line 192 in section 4.1
* The sentence at the end of line 199 in Section 4.1 is grammatically incorrect: "And the quality for $s_o$ could get by" should be something like "The quality of $s_o$ is given by" or something like that?
* The sentence spanning lines 203 and 204: "In the case of using the same similarity calculation method, since the performance of this evaluation approach depends on the orthogonality of the ASR models’ performance." is grammatically incomplete, it should probably say something like "This metric requires that the ASR models' performance be uncorrelated with each other so that we do not see consistent ASR errors on the same input."
* Missing spaces around the period on line 212/213 in section 4.2
* Missing space after "Section 4.2." on line 261 in Section 4.3
* Missing space after the comma on line 283 in Section 5.1
* Caption for Table 1 should state what Num is instead of just stating the names of the columns.
* Missing space on line 308 between "Table 2" and "provides"
* On line 352, "ROUGEL" should be "ROUGE-L"
* On line 406, "fintinue" should be "finetune"
* Caption for Table 8 is just wrong, it isn't "Examples of speech style descriptions generated by GPT-4", it seems like someone copy and pasted the caption for Table 2 and didn't edit it afterwards.
* In Section E.1, License is spelled "Lince" many times
* Extra period on line 335/336 "alignment.,"
* Missing spaces "thenear" on line 545 in the Reproducibility statement and "thetransparency" on line 555/556 in the Ethics and Ethics Statement. Also, the title of the Ethics statement is "Ethics and Ethics Statement", shouldn't this just be "Ethics statement"?

---

> ### Author Response · Authors · 2024-11-25
> **Response by Authors to Reviewer 2kw3 (Part.1)**
>
> We are deeply grateful for your recognition of our work's innovation and thoroughness, as well as their constructive feedback. We have addressed each suggestion on the manuscript's weaknesses and made the necessary revisions.
>
> > Q1: One of the main issues with this paper is ...
>
> We report additional test results. We use different threshold settings to filter usable training samples and synthesize the test set with MeloTTS.
>
> All test sets were validated for linguistic information using an ASR model, and any test samples that failed the validation were discarded to ensure the quality of the test set.
>
> We use ROUGE-L as a metric to evaluate the inference results. The results are shown in the table below.]()
>
>
> | Threshold 	 | Training   Target 	 |   Drop  	    |   Quoref 	   |   Ropes 	    | NarrativeQA 	  |  Average 	   |
> |:-----------:|:-------------------:|:------------:|:------------:|:------------:|:--------------:|:------------:|
> |   -     	   |     -         	     |   17.40  	   |   55.98  	   |   42.69  	   |   43.02    	   |   39.77  	   |
> | 0.00     	  |  LLM Continue   	   |   30.08  	   |   75.05  	   |   57.15  	   |   47.88    	   |   52.54  	   |
> |  0.00    	  |    Golden      	    |   42.78  	   |   86.58  	   |   60.48  	   |   52.15    	   |   60.50  	   |
> |  85.00   	  |    Golden      	    |   42.95  	   |   85.18  	   |   56.47  	   |   53.61    	   |   59.55  	   |
> |  90.00   	  |    Golden      	    | **44.35**  	 | **86.81**  	 | **64.24**  	 | **56.76**    	 | **68.43**  	 |
> |  95.00   	  |    Golden      	    |   41.86  	   |   86.73  	   |   58.55  	   |   54.61    	   |   60.44  	   |
>
> *Threshold* refers to the minimum quality score for valid training set samples, with the quality score calculated according to Equation (5).
>
>
> > Q2: Parts of this manuscript are poorly written and explained...
>
> We use ROUGE-L to evaluate the model’s performance on NarrativeQA.
>
> > Q3: The implementation details in section 5.1 on page 7 also state that top-p for generation is 10 (I am assuming 10%), which is extremely low, this might as well be greedy sampling.
>
> We carefully reviewed our code and found that during inference, we used the default settings provided by Qwen2-Audio-7B-Instruct rather than the default settings of the inference library.
> The specific settings are shown below, and we will correct these settings in the revised manuscript.
>
> ```json
> {
>   "chat_format": "chatml",
>   "eos_token_id": [151643,151645],
>   "pad_token_id": 151643,
>   "do_sample": true,
>   "top_k": 20,
>   "top_p": 0.5,
>   "temperature": 0.7,
>   "repetition_penalty": 1.1,
>   "transformers_version": "4.38.1"
> }
> ```
>
> We will revise the settings in the new manuscript to:
>
> For the generative evaluation phase, we evaluated our results on a single NVIDIA A40 GPU, setting top-p to 0.5, top-k to 20, repetition penalty to $1.1$ and temperature to 0.7.
>
>
> > Q4: Furthermore the use of Knowledge Fusion is not very well...
>
> The proposed synthesis framework is a two-stage framework, with the first stage illustrated in Figure 2 and the second stage in Figure 3.
>
> During the knowledge fusion stage, we used LoRA to train separate rewriting models for each dataset to rewrite the queries that were not successfully rewritten in the first stage.
> The training data consisted of samples successfully rewritten in the first stage. The criteria for selecting these successful samples are detailed in Section 4.3.
>
>
> > Q5: Furthermore, the abstract, introduction and conclusion all repeat this statement....
>
> The usability calculation results are derived from the mean *PASS* scores in the experiments, which are distributed across three tables in Tabel 5.
> To clarify the source of these results, we have consolidated this metric into the table below. We will include the following table in the appendix of the revised PDF.
>
>
> |  Method   	   | DROP  	  | Quoref 	 | ROPES 	  | NarrativeQA 	 | TAT-QA 	 | SQUAD1.1 	 | SQUAD2.0 	 | Average 	 |
> |:-------------:|:--------:|:--------:|:--------:|:-------------:|:--------:|:----------:|:----------:|:---------:|
> |  Original  	  | 74.40  	 | 84.78  	 | 89.80  	 |  83.63    	   | 25.85  	 | 73.39   	  | 73.45   	  | 72.19   	 |
> |   Phi3    	   | 89.82  	 | 92.69  	 | 94.60  	 |  91.04    	   | 79.46  	 | 73.39   	  | 73.45   	  | 84.92   	 |
> |  Qwen2    	   | 87.28  	 | 91.70  	 | 94.09  	 |  89.38    	   | 56.63  	 | 83.46   	  | 83.53   	  | 83.72   	 |
> |  Llama3   	   | 90.40  	 | 92.23  	 | 94.44  	 |  89.37    	   | 80.21  	 | 85.91   	  | 85.96   	  | 88.36   	 |
> | Ours w/o KF 	 | 93.43  	 | 95.24  	 | 96.32  	 |  93.54    	   | 88.72  	 | 90.19   	  | 90.19   	  | 92.52   	 |
> |   Ours    	   | 94.11  	 | 95.59  	 | 96.71  	 |  94.29    	   | 89.12  	 | 90.78   	  | 90.93   	  | 93.07   	 |

---

> > ### Author Response · Authors · 2024-11-25
> > **Response by Authors to Reviewer 2kw3 (Part.2)**
> >
> > > Q6: Ablations are incomplete. In particular, it is unclear whether this technique improves over just having the LLMs rewrite the queries and using raw TTS after that.
> >
> > We present the experimental results of using a single LLM for rewriting in Table 5.
> > The results demonstrate that our proposed method outperforms simply having the LLM rewrite the queries and then using raw TTS afterward.
> >
> > > Q7: On line 053, you say that these training methods "treat the speech modality as files rather than instructions", what does this mean?
> >
> > “Treat the speech modality as files rather than instructions” means that during the training phase, the model treats speech input as context rather than commands to be followed.
> > Without additional textual instructions, the model defaults to performing automatic speech recognition.
> > This phenomenon is also widely observed in vision-language models, where, when only an image is input, the model defaults to providing a description of the image.
> >
> >
> > > Q8: ...Where is any reference for this phenomenon?
> >
> > We provide some examples below, which are derived from Qwen-Audio-7B and Qwen2-Audio-7B. These models were trained with a large number of ASR tasks and other classic speech tasks.
> > Upon receiving voice command inputs, they ignored the voice commands and directly performed automatic speech recognition instead of responding according to the voice commands.
> >
> > Sample 1
> >
> > ```text
> > Speech:Is Bob more or less likely to be having his body respond to his current temperature than Gretchen?
> >
> > Qwen2-Audio-7B Output:Is Bob more or less likely to be having his body respond to his current temperature than Gretchen?
> >
> > Qwen-Audio-7B Output:Is Bob more or less likely to be having his body respond to his current temperature than Gretchen?
> > ```
> >
> > Sample 2
> > ```text
> > Speech:Which of the two most likely has their blood and sweat vessels dilated?
> >
> > Qwen2-Audio-7B Output:Which of the two most likely has their blood and sweat vessels dilated?
> >
> > Qwen-Audio-7B Output:Which of the two most likely has their blood and sweat vessels dilated?
> > ```
> >
> >
> > > Q9: In Figure 1, it is stated that...
> >
> > The same sentence, ending with different punctuation marks,
> > can convey completely different meanings. Below, we present a more suitable example.
> > The original sentence expresses a question, while the ASR-recognized sentence, lacking proper punctuation, conveys a tone of disdain or assertion.
> > ```text
> > Original: What do you know?
> >
> > ASR: What do you know
> >
> > ```

---

> > > ### Comment · Reviewer_2kw3 · 2024-11-26
> > >
> > > I appreciate the authors' responses to my concerns:
> > >
> > > 1. Regarding the response to Q1, now I am more concerned that there is this new threshold variable that has been introduced in the existing table. The existing Table 4 in the paper did not have this parameter, which adds to my concern, that the experimental setup is not complete in the paper. For instance, on line 408-409, it just says that these results are "based on different experimental setups and query modalities", but these exact setups are not mentioned anywhere. These experiments should have a proper experimental setup section in the paper (even in the appendix) to properly document the results.
> > > 2. Regarding the response to Q8, I was talking about the results in Table 4 (and the additional results in your response). The only comparison between your technique and a baseline is the "Direct use TTS" method, but I would like to see a proper ablation where you use a TTS model on-top of an query that a single LLM has rewritten. Table 5 only shows some quality evaluation results, but I am wondering about these results on the downstream task itself.
> > > 3. Overall, I still see lots of errors and typos in the main text, will these be cleaned up in the final manuscript?

---

> > > > ### Author Response · Authors · 2024-11-27
> > > > **Response by Authors to Reviewer 2kw3 (Part.3)**
> > > >
> > > > Thank you very much for your response. We value your insights, and in response to your points of concern, we offer the following explanations:
> > > >
> > > > > Regarding the response to Q1, now I am more concerned that there is...
> > > >
> > > > This threshold is used to control the sampling of the training set. We have updated our manuscript and provided an explanation in Section 5.1, Dataset.
> > > >
> > > > The explanation is as follows.
> > > >
> > > > ```text
> > > > For the training data to train LSLMs, we calculate the data quality using Equation (5) and set a threshold t. The quality of all data used for training the model must exceed t.
> > > > For data derived from the same original text, only the speech instruction with the highest quality that meets the threshold requirement is included in the training.
> > > > ```
> > > >
> > > > We have provided detailed experimental settings in Appendix B.1 of the revised manuscript.
> > > >
> > > >
> > > >
> > > > > Regarding the response to Q8, I was talking about the results in Table 4 (and the additional results in your response). The only comparison between your technique and a baseline is the "Direct use TTS" method, but I would like to see a proper ablation where you use a TTS model on-top of an query that a single LLM has rewritten. Table 5 only shows some quality evaluation results, but I am wondering about these results on the downstream task itself.
> > > >
> > > > We have added more detailed ablation results, which were conducted under the multi-speaker setting.
> > > >
> > > > In the ablation results, a slight performance degradation occurs on some data when only LLM query rewriting is applied.
> > > > This is primarily due to instances where the LLM sometimes fails to follow instructions during query rewriting, resulting in formatting issues.
> > > >
> > > > The method settings are as follows.
> > > >
> > > > (1) Phi3/Qwen2/Llama3: Follow the synthesis framework in Figure 2 but only use Phi-3-small-8k-instruct/Qwen2-7B-Instruct/Llama-3-8B-Instruct to rewrite queries.
> > > > (2) Phi3/Qwen2/Llama3 w/o Original: Follow the synthesis framework in Figure 2 but only use Phi-3-small-8k-instruct/Qwen2-7B-Instruct/Llama-3-8B-Instruct to rewrite queries and not use original text.
> > > > (2) Ours w/o KF: Use only the synthesis framework in Figure 2 without knowledge fusion.
> > > >
> > > > | Method              	 | ASR Model 	 | DROP  	  | Quoref 	 | ROPES 	  | NarrativeQA 	 | TAT-QA 	 | squad1.1 	 | squad2.0 	 | Average 	 |
> > > > |-----------------------|-------------|:--------:|:--------:|:--------:|:-------------:|:--------:|:----------:|:----------:|-----------|
> > > > | Original            	 | Ours      	 | 93.71  	 | 96.25  	 | 97.07  	 |  95.87    	   | 82.24  	 | 93.40   	  | 93.42   	  | 93.14   	 |
> > > > | Phi3                	 | Ours      	 | 97.24  	 | 98.07  	 | 98.32  	 |  97.64    	   | 95.29  	 | 96.39   	  | 96.39   	  | 97.05   	 |
> > > > | Phi3 w/o Original   	 | Ours      	 | 95.11  	 | 95.84  	 | 95.75  	 |  94.90    	   | 94.09  	 | 93.65   	  | 93.60   	  | 94.70   	 |
> > > > | Llama3              	 | Ours      	 | 97.30  	 | 97.88  	 | 98.25  	 |  97.22    	   | 95.27  	 | 96.35   	  | 96.34   	  | 96.94   	 |
> > > > | Llama3 w/o Original 	 | Ours      	 | 93.91  	 | 93.33  	 | 94.62  	 |  92.70    	   | 93.66  	 | 92.95   	  | 92.92   	  | 93.44   	 |
> > > > | Qwen2               	 | Ours      	 | 96.45  	 | 97.64  	 | 98.05  	 |  97.08    	   | 90.31  	 | 95.68   	  | 95.69   	  | 95.84   	 |
> > > > | Qwen2 w/o Oringal   	 | Ours      	 | 92.01  	 | 92.22  	 | 93.00  	 |  92.31    	   | 89.07  	 | 92.00   	  | 92.46   	  | 91.87   	 |
> > > > | Ours w/o KF         	 | Ours      	 | 98.02  	 | 98.57  	 | 98.79  	 |  98.14    	   | 97.12  	 | 97.37   	  | 97.36   	  | 97.91   	 |
> > > > | Ours                	 | Ours      	 | 98.11  	 | 98.62  	 | 98.82  	 |  98.24    	   | 97.18  	 | 97.47   	  | 97.49   	  | 97.99   	 |
> > > >
> > > > > Overall, I still see lots of errors and typos in the main text, will these be cleaned up in the final manuscript?
> > > >
> > > > We sincerely thank you for your constructive feedback. We have addressed these issues in the revised manuscript.

---

> > > > > ### Comment · Reviewer_2kw3 · 2024-11-27
> > > > >
> > > > > I appreciate the authors' additional experiments and responses. Overall, I am impressed by the sheer number of experiments and results that the authors have included in the current version of the manuscript for their ablations. I am willing to raise the score 3 -> 5 if the following concerns can be addressed:
> > > > >
> > > > > * Regarding your experimental setup, it is still a little confusing to me the exact difference between Golden and LLM Continue. How exactly are these alignment targets?
> > > > > * In the first row of Table 4, what exactly is that experiment that has no Training target or threshold or construction method?
> > > > > * There also seems to not be a text normalization ablation in Table 4 either.
> > > > >
> > > > > Beyond this, I have some concerns about the scope of this paper:
> > > > > * Regarding Q9, while preserving this kind of paralinguistic content can be important, it is unclear to me whether these datasets really benefit from this kind of information being preserved?
> > > > > * It is noted in the paper on lines 280-283 that `In real user interaction scenarios, it is rare to use lengthy speech to provide a detailed task definition; instead, users often ask brief questions expecting responses that meet their expectations. Therefore, in this paper, we selected several QA datasets with short questions to validate the effectiveness of our method.` One may be concerned that this decision may have favored the efficacy of your technique since there is less content overall, so small mistakes may heavily impact the performance of the model downstream.
> > > > >
> > > > > Side note: Your manuscript appears to still have some typos:
> > > > > * On line 293: Baseline and Aboluetion Method, no idea what "Aboluetion" is, is this supposed to be Ablation? In which case, I think that just Baselines would suffice?
> > > > > * On line 545-546, thenear -> the near

---

> ### Author Response · Authors · 2024-11-27
> **Response by Authors to Reviewer 2kw3 (Part.4)**
>
> Thank you very much for your thoughtful and detailed response. We value your insights, and in response to your points of concern, we offer the following explanations:
>
> > Regarding your experimental setup, it is still a little confusing to me the exact difference between Golden and LLM Continue. How exactly are these alignment targets?
>
> Golden refers to high-quality human-provided answers corresponding to the speech instructions, while Continue refers to answers generated by the LLM.
>
> Limited by the capabilities of the LLM itself, the answers it generates often differ from the true answers and can even be misleading.
> This increases the difficulty of model learning and may introduce incorrect training objectives.
>
> Constrained by the inherent limitations of the LLM, the answers it generates may deviate from the true answers and even be misleading.
> This can increase the difficulty of model learning and potentially result in incorrect training objectives.
>
> We present the performance of LLM-generated answers on the training set below to quantify this gap. The results are evaluated using ROUGE-L.
>
> | Dataset     	 | Continue 	 | Golden 	 |
> |---------------|------------|----------|
> | drop        	 | 58.34    	 | 100    	 |
> | narrativeqa 	 | 70.15    	 | 100    	 |
> | quoref      	 | 75.48    	 | 100    	 |
> | ropes       	 | 17.00    	 | 100    	 |
>
>
> For the alignment method, we use the cross-entropy loss function to calculate the loss, which is computed only for the Response part in Table 11. The instruction templates used during the training process are provided in Table 11.
>
> > In the first row of Table 4, what exactly is that experiment that has no Training target or threshold or construction method?
>
> The first row of Table 4 shows the evaluation results of the untrained backbone model, Qwen2-Audio-7B-Instruct. We have updated the table in the revised manuscript for better readability.
>
> > There also seems to not be a text normalization ablation in Table 4 either.
>
> Table 4 presents the evaluation results on downstream tasks after training the model with the speech instruction data we constructed. TN, as a baseline data synthesis method, has its results shown in Tables 2, 3, 14, and 15.
> It is significantly inferior to our method in terms of data synthesis quality and underperforms in data usability (TN 82.05%, Ours 93.07%).
>
> > It is noted in the paper on lines 280-283 that
>
> To demonstrate the effectiveness of our method on longer texts, we separately report the performance of various methods on samples with original text lengths exceeding 20 words in each dataset.
> We use SIM as the evaluation metric under the single-speaker setting. The experimental results show that our method maintains good performance on longer texts.
>
> |   TTS Model      	    | Method   	       | Drop  	  | narrativeqa 	 | Quoref 	 | ropes 	  | squad1.1 	 | squad2.0 	 | tatqa 	  | Average 	 |
> |:---------------------:|------------------|:--------:|:-------------:|:--------:|:--------:|:----------:|:----------:|:--------:|:---------:|
> | parler-tts-large-v1 	 | Original 	       | 89.94  	 |  95.50    	   | 94.02  	 | 95.03  	 | 93.77   	  | 93.39   	  | 86.00  	 | 92.52  	  |
> |           	           | TN       	       | 94.02  	 |  97.80    	   | 96.75  	 | 97.71  	 | 96.53   	  | 96.32   	  | 91.53  	 | 95.81  	  |
> |           	           | Phi3     	       | 94.31  	 |  97.91    	   | 97.24  	 | 97.67  	 | 96.96   	  | 96.73   	  | 91.45  	 | 96.04  	  |
> |           	           | Qwen2    	       | 94.04  	 |  97.73    	   | 97.06  	 | 97.70  	 | 97.09   	  | 95.91   	  | 91.14  	 | 95.81  	  |
> |           	           | Llama3   	       | 94.36  	 |  97.92    	   | 97.01  	 | 97.56  	 | 93.77   	  | 96.83   	  | 92.99  	 | 95.78  	  |
> |           	           | Ours w/o KF    	 | 95.83  	 |  98.64    	   | 98.11  	 | 98.33  	 | 97.61   	  | 97.57   	  | 94.50  	 | 97.23  	  |
>
>
> > Regarding Q9, while preserving this kind of paralinguistic content can be important, it is unclear to me whether these datasets really benefit from this kind of information being preserved?
>
> Since the embedding model considers punctuation when calculating similarity, reducing this influence helps improve the accuracy of quality assessment.
>
> In the examples below, slight changes in similarity can be observed as punctuation changes.
>
> In our data, due to the high average similarity between the linguistic information of synthesized speech and the original text, variations in punctuation can also affect the evaluation results.
>
> ```text
> what do you know?
> SIM:1.000
> what do you know
> SIM:0.993
> what do you know.
> SIM:0.987
> what do you know!
> SIM:0.971
> ```

---

> ### Comment · Reviewer_2kw3 · 2024-11-29
>
> I thank the authors for their detailed responses.
>
> My main point in adding that question about the paralinguistic content leads me to one of my main concerns: a majority of this paper is related to showing that this technique improves WER/SIM/PASS on the synthetically generated data, not with downstream tasks, which ultimately is the main goal. Time permitting, I would mainly like to see some kind of analysis that these metrics are strongly correlated with downstream performance, or some references to this phenomena. Perhaps some kind of correlation analysis or something to show that these examples can be a strong proxy/indicator for downstream performance.
>
> I am raising my soundness and presentation scores accordingly, but I would need to see convincing evidence addressing my concern to raise my overall score.

---

> ### Author Response · Authors · 2024-11-30
> **Response by Authors to Reviewer 2kw3 (Part.5)**
>
> Thank you very much for your thoughtful and detailed response. We value your insights, and in response to your points of concern, we offer the following explanations:
>
> In this paper, we assess data quality primarily based on the SIM metric, where Pass refers to the proportion of synthetic data with a SIM greater than 0.9.
> To verify the correlation between this metric and the performance of downstream tasks, we include the following experiments:
>
> Specifically, we use the best ASR results corresponding to the speech data as the query, with the same instruction template, and use Llama3 to generate responses.
> This eliminates the impact of synthetic speech training data of varying quality on the experimental results. We mainly compare the experimental results under the following settings.
>
> Ref: Directly using the original text as the query
>
> Ours: The best ASR recognition results of the synthetic speech data generated by our method in a multi-speaker setting.
>
> TN: The best ASR recognition results of the synthetic speech data generated by the text normalization method in a multi-speaker setting.
>
> Original: The best ASR recognition results of the synthetic speech data generated by the original method in a multi-speaker setting.
>
> The best ASR recognition results are obtained by selecting the one with the highest similarity to the original text.
>
> We report the results on the training sets of Drop, Quoref, and Ropes.
>
> The experimental results show that as the average SIM increases, there is a consistent improvement in performance on downstream tasks.
>
> |  Method  |  Drop | Quoref | Ropes | Average |   SIM  |
> |:--------:|:-----:|:------:|:-----:|:-------:|:------:|
> |   Ref    | 58.34 |  75.48 | 17.00 |  50.27  | 100.00 |
> |   Ours   | 56.78 |  74.64 | 16.68 |  49.37  |  98.51 |
> |    TN    | 56.29 |  73.92 | 16.61 |  48.94  |  96.91 |
> | Original | 55.41 |  73.56 | 16.34 |  48.44  |  95.67 |

---

> > ### Comment · Reviewer_2kw3 · 2024-12-02
> >
> > I thank the authors for adding these experiments on such short notice. Indeed, this seems to show some correlation between SIM and performance. Please include these experiments in the final version of your manuscript. I have raised my score 3 -> 5.

---

> ### Author Response · Authors · 2024-12-04
>
> Dear Reviewer 2kw3,
>
> Thank you for your positive response and support for our work! We noticed that the rating has not been updated, so we would like to confirm this with you.
>
> Thank you again for your time and assistance!

---

### Official Review · Reviewer_TuG5 · 2024-11-04

**Soundness:** 3
**Presentation:** 3
**Contribution:** 3
**Rating:** 6
**Confidence:** 3

**Summary:**

This paper presents a novel framework for constructing high-quality speech instruction datasets using query rewriting with multi-LLM knowledge fusion. The key contributions are:

- A query rewriting framework that leverages multiple LLMs to rewrite text instructions to better match TTS model training distributions while preserving semantics.
- A multi-agent annotation and validation approach using multiple ASR models and embedding-based similarity metrics to ensure quality.
- A knowledge fusion method that combines strengths of different LLMs to handle challenging rewriting cases.

**Strengths:**

- Novel and practical solution to an important problem in speech instruction dataset creation.
- Comprehensive evaluation across multiple datasets and metrics.
- Detailed ablation studies validating each component.
- Cost-effective compared to human annotation.
- Strong experimental results showing clear improvements in both data quality and downstream task performance.
- Good technical novelty in combining multiple LLMs and agents for robust performance.

**Weaknesses:**

- Limited analysis of failure cases and error patterns.
- No direct comparison with other query rewriting methods from adjacent domains.
- Validation relies heavily on embedding similarity - could benefit from human evaluation.
- Parameter sensitivity analysis missing (e.g., impact of different thresholds).
- Scalability and computational costs not thoroughly discussed.

**Questions:**

/

---

> ### Author Response · Authors · 2024-11-25
> **Response by Authors to Reviewer TuG5 (Part.1)**
>
> We deeply appreciate the time and effort you have dedicated to reviewing this paper. We value your insights, and in response to your points of concern, we offer the following explanations:
>
> > Q1: No direct comparison with other query rewriting methods from adjacent domains.
>
> The work most closely related to ours is text normalization, a common industrial approach used to optimize the input for TTS models, which rewriting queries to fit the input requirements of the TTS model without altering their expression.
> We added a text normalization method as a baseline for comparison, which achieved 98% accuracy in the Google Text Normalization Challenge.
>
> To demonstrate the generality of our method and control the influence of speaker style descriptions on the synthesis results, the TTS models in the following experiments use a single speaker style setting during synthesis.
>
> For Parler-TTS-Large-v1, we use *Jon's voice is monotone yet slightly fast in delivery, with a very close recording that almost has no background noise* as the speaker style descriptions.
>
> For MeloTTS, we use the officially provided ‘EM-US’ setting.
>
> For mms-tts-eng, no additional speaker style control measures are provided by the official implementation, so we follow the default settings.
>
> The experimental results are shown below, where we use SIM as a metric to evaluate the results.
>
> |   TTS   Model     	   | Method   	          |  Drop    	  | narrativeqa 	 | Quoref   	  |  ropes   	  | squad1.1  	 | squad2.0  	 |  tatqa   	  | Average  	  |
> |:---------------------:|---------------------|:-----------:|:-------------:|:-----------:|:-----------:|:-----------:|:-----------:|:-----------:|:-----------:|
> |    MeloTTS       	    | Original 	          | 96.61    	  |  95.13    	   | 97.27    	  | 98.22    	  | 95.75    	  | 93.48    	  | 97.43    	  | 96.27    	  |
> |           	           | TN       	          | 97.50    	  |  96.36    	   | 98.00    	  | 98.83    	  | 94.86    	  | 95.54    	  | 97.96    	  | 97.01    	  |
> |           	           | Phi3     	          | 97.57    	  |  95.82    	   | 97.81    	  | 98.70    	  | 96.74    	  | 95.71    	  | 98.29    	  | 97.23    	  |
> |           	           | Qwen2    	          | 97.59    	  |  96.42    	   | 98.01    	  | 98.85    	  | 96.98    	  | 95.68    	  | 98.19    	  | 97.39    	  |
> |           	           | Llama3   	          | 97.65    	  |  96.32    	   | 98.01    	  | 98.91    	  | 97.10    	  | 95.78    	  | 98.30    	  | 97.44    	  |
> |           	           | Ours  w/o KF   	    | **98.19** 	 |  **97.02** 	  | **98.37** 	 | **99.12** 	 | **97.59** 	 | **96.80** 	 | **98.53** 	 | **97.94** 	 |
> |   mms-tts-eng     	   | Original 	          | 92.05    	  |  91.46    	   | 93.77    	  | 93.42    	  | 91.28    	  | 91.23    	  | 84.61    	  | 91.12    	  |
> |           	           | TN       	          | 94.22    	  |  **93.82** 	  | 95.64    	  | 95.14    	  | 93.86    	  | 93.26    	  | 90.75    	  | 93.81    	  |
> |           	           | Phi3     	          | 95.06    	  |  93.00    	   | 95.46    	  | 94.97    	  | 93.90    	  | 93.86    	  | 92.52    	  | 94.11    	  |
> |           	           | Qwen2    	          | 95.18    	  |  91.47    	   | 95.51    	  | 95.04    	  | 94.32    	  | 94.62    	  | 91.81    	  | 93.99    	  |
> |           	           | Llama3   	          | 94.52    	  |  91.47    	   | 95.51    	  | 94.98    	  | 94.85    	  | 94.82    	  | 94.30    	  | 94.35    	  |
> |           	           | Ours  w/o KF      	 | **96.86** 	 |  93.01    	   | **96.51** 	 | **95.95** 	 | **95.96** 	 | **96.07** 	 | **96.17** 	 | **95.79** 	 |
> | Parler-TTS-Large-v1 	 | Original 	          | 93.04    	  |  95.26    	   | 95.51    	  | 96.01    	  | 92.75    	  | 91.99    	  | 81.63    	  | 92.31    	  |
> |           	           | TN       	          | 96.41    	  |  97.39    	   | 97.63    	  | 98.06    	  | 95.89    	  | 95.61    	  | 89.07    	  | 95.72    	  |
> |           	           | Phi3     	          | 97.21    	  |  97.40    	   | 97.95    	  | 97.97    	  | 96.49    	  | 96.19    	  | 93.80    	  | 96.72    	  |
> |           	           | Qwen2    	          | 97.17    	  |  97.51    	   | 97.93    	  | 98.05    	  | 96.29    	  | 95.64    	  | 91.63    	  | 96.32    	  |
> |           	           | Llama3   	          | 97.41    	  |  97.41    	   | 97.78    	  | 96.74    	  | 96.74    	  | 96.44    	  | 96.01    	  | 96.93    	  |
> |           	           | Ours  w/o KF     	  | **98.09** 	 |  **98.19** 	  | **98.52** 	 | **98.30** 	 | **97.31** 	 | **97.32** 	 | **97.21** 	 | **97.85** 	 |
>
> Which *TN* mean the text normalization, *Phi3* mean only use Phi-3-small-8k-instruct to rewrite query, *Qwen2* mean only use Qwen2-7B-Instruct to rewrite query, *Llama3* mean only use Llama-3-8B-Instructt to rewrite query,
> and *Ours  w/o KF* refers to using only the training-free rewriting framework illustrated in Figure 1.

---

> > ### Author Response · Authors · 2024-11-25
> > **Response by Authors to Reviewer TuG5 (Part.2)**
> >
> > > Q2:Limited analysis of failure cases and error patterns.
> >
> >
> > We provide more examples below to demonstrate the effectiveness of our proposed method.
> > In Case 3, we demonstrate the advantage of query rewriting over text normalization,
> > namely reorganizing sentences to make them more aligned with natural spoken expression without altering their original meaning.
> >
> > **Sample 1**
> > * Original Text:Which groups in percent are larger than 17%?
> > * Llama-3-8B-Instruct Rewrite:Which groups in percentage are larger than seventeen percent?[✓]
> > * Phi-3-small-8k-instruct Rewrite:Which groups in percent are larger than seventeen percent?[✓]
> > * Qwen2-7B-Instruct Rewrite:Which groups in percent are larger than 17%?
> >
> >
> > **Sample 2**
> > * Original Text::Who caught the longest TD pass?
> > * Llama-3-8B-Instruct Rewrite:Who caught the longest touch-down pass?
> > * Phi-3-small-8k-instruct Rewrite:Who caught the longest touchdown pass?[✓]
> > * Qwen2-7B-Instruct Rewrite:Who caught the longest TD pass?
> >
> > **Sample 3**
> > * Original Text: Were there more households with female householder (with no husband present) or male household (with no wife present)?
> > * Llama-3-8B-Instruct Rewrite:Were there more households with female householder which with no husband present or male household which with no wife present?[✓]
> > * Phi-3-small-8k-instruct Rewrite:Were there more households with female householder (with no husband present) or male household (with no wife present)?
> > * Qwen2-7B-Instruct Rewrite:Were there more households with female householder (with no husband present) or male household (with no wife present)?
> >
> >
> >
> > > Q3: Validation relies heavily on embedding similarity - could benefit from human evaluation.
> >
> > Thank you for your suggestion.
> > To more thoroughly evaluate the semantic consistency between our rewritten content and the original, we employed a voting mechanism using multiple LLMs to simulate human annotation.
> > This approach verifies the consistency of linguistic information between the synthesized speech generated from the rewritten queries and the original text.
> >
> > Specifically, we used Qwen2-7B-Instruct, Llama-3-8B-Instruct, and Phi-3-small-8k-instruct to validate consistency, employing the following prompts to guide the models in providing judgment results.
> > We conducted tests on the Drop, Quoref, ropes, and tatqa datasets, utilizing Parler-TTS-Large-v1 as TTS Model.
> >
> >
> > ```text
> > # Task #
> >
> > Please evaluate whether the following two sentences convey identical meanings in content. Use yes or no to give your verdict.
> >
> > # Sentence 1 #
> >
> > {original_text}
> >
> > # Sentence 2 #
> >
> > {rewrite_text}
> > ```
> > *original_text* refers to the original text, while *rewrite_text* refers to the linguistic information of the speech synthesized via TTS after rewriting, which is obtained through an ASR model.
> >
> >
> > The following are the experimental results. We use the pass rate as the metric, where a sentence is considered to pass the validation if more than two models deem it consistent.
> >
> >
> > | Method   	         |   Drop  	    |   tatqa 	    |  Quoref 	   |   ropes 	    |  Average 	   |
> > |--------------------|:------------:|:------------:|:-----------:|:------------:|:------------:|
> > | Original 	         |   71.20  	   |   22.82  	   |  83.84  	   |   78.28  	   |   64.03  	   |
> > | TN       	         |   85.37  	   |   41.41  	   |  92.09  	   |   87.75  	   |   76.66  	   |
> > | Phi3     	         |   92.98  	   |   77.58  	   |  95.31  	   |   89.27  	   |   88.79  	   |
> > | Qwen2    	         |   93.74  	   |   77.45  	   |  95.60  	   |   90.49  	   |   89.32  	   |
> > | Llama3   	         |   94.49  	   |   79.61  	   |  93.96  	   |   87.38  	   |   88.86  	   |
> > | Ours    w/o KF   	 |   97.42  	   |   97.41  	   |  97.89  	   |   94.02  	   |   96.69  	   |
> > | Ours       	       | **98.15**  	 | **97.92**  	 | **98.23** 	 | **95.06**  	 | **97.16**  	 |
> > Which *TN* mean the text normalization, *Phi3* mean only use Phi-3-small-8k-instruct to rewrite query, *Qwen2* mean only use Qwen2-7B-Instruct to rewrite query, *Llama3* mean only use Llama-3-8B-Instructt to rewrite query, and *Ours  w/o KF* refers to using only the training-free rewriting framework illustrated in Figure 1.

---

> ### Author Response · Authors · 2024-11-25
> **Response by Authors to Reviewer TuG5 (Part.3)**
>
> > Q4: Parameter sensitivity analysis missing (e.g., impact of different thresholds).
>
> We use different threshold settings to filter usable training samples and synthesize the test set with Melo to validate the correctness of our threshold selection.
>
> All test sets were validated for linguistic information using an ASR model, and any test samples that failed the validation were discarded to ensure the quality of the test set.
>
> We use ROUGE-L as a metric to evaluate the inference results. The results are shown in the table below.
>
> | Training   Target 	 | Threshold 	 |   Drop  	    |   Quoref 	   |   Ropes 	    | NarrativeQA 	  |  Average 	   |
> |:-------------------:|:-----------:|:------------:|:------------:|:------------:|:--------------:|:------------:|
> |     -         	     |   -     	   |   17.40  	   |   55.98  	   |   42.69  	   |   43.02    	   |   39.77  	   |
> |  LLM Continue   	   | 0.00     	  |   30.08  	   |   75.05  	   |   57.15  	   |   47.88    	   |   52.54  	   |
> |    Golden      	    |  0.00    	  |   42.78  	   |   86.58  	   |   60.48  	   |   52.15    	   |   60.50  	   |
> |    Golden      	    |  85.00   	  |   42.95  	   |   85.18  	   |   56.47  	   |   53.61    	   |   59.55  	   |
> |    Golden      	    |  90.00   	  | **44.35**  	 | **86.81**  	 | **64.24**  	 | **56.76**    	 | **68.43**  	 |
> |    Golden      	    |  95.00   	  |   41.86  	   |   86.73  	   |   58.55  	   |   54.61    	   |   60.44  	   |
>
>
> The experimental results indicate that using 0.9 as the threshold for training set sampling demonstrates consistently good performance on downstream tasks.
>
> > Q5: Scalability and computational costs not thoroughly discussed.
>
> ### Scalability
>
> We provide a detailed report in Q2 on the performance of our method across multiple datasets and TTS models.
> Our method demonstrates consistently good performance on different TTS models, indicating its scalability.
>
>
> ### computational costs
>
> We have detailed the costs associated with the proposed speech instruction construction method and added a discussion on the costs of MeloTTS, a vocoder-based TTS model.
>
> We primarily discussed the cost issues of several different speech synthesis methods and provided the costs of different speech synthesis methods across various datasets.
>
> * MeloTTS/Parler+Original:Synthesizes speech using MeloTTS/Parler-TTS-Large-v1, with only the original text as input to the TTS model.
> * MeloTTS/Parler+Ours w/o KF:Synthesizes speech using MeloTTS/Parler-TTS-Large-v1, following the method illustrated in Figure 1.
> * Parler+Ours:Synthesizes speech using Parler-TTS-Large-v1, following the method illustrated in Figure 1, while applying the knowledge fusion method shown in Figure 2 to rewrite queries that failed in the first stage.~~
>
>
>
> | Speech   Collection Method | Quality Validation | Speakers Num | Human Time Cost | Gpu Time Cost | Money Cost |
> |:--------------------------:|:------------------:|:------------:|:---------------:|:-------------:|:----------:|
> |           Human            |       Human        |      ∞       |       562       |       0       |    4215    |
> |           Human            |     Single ASR     |      ∞       |       281       |       6       |  2110.02   |
> |           Human            |  Multi ASR + Emb   |      ∞       |       281       |      16       |  2114.22   |
> |      MeloTTS+Original      |     Single ASR     |      1       |        0        |      14       |    5.88    |
> |      Parler+Original       |     Single ASR     |     192      |        0        |      127      |   53.34    |
> |    MeloTTS+Ours w/o KF     |  Multi ASR + Emb   |      1       |        0        |      82       |   34.44    |
> |     Parler+Ours w/o KF     |  Multi ASR + Emb   |     192      |        0        |      534      |   224.28   |
> |        Parler+Ours         |  Multi ASR + Emb   |     192      |        0        |      558      |   234.36   |
>
>
> |   Dataset   | MeloTTS | Parler | MeloTTS+Ours w/o KF | Parler+Ours w/o KF | Parler+Ours |
> |:-----------:|:-------:|:------:|:-------------------:|:------------------:|:-----------:|
> |    Drop     |   0.5   |   14   |          2          |         56         |     61      |
> |    ROPES    |   0.5   |   7    |          2          |         28         |     31      |
> |   Quoref    |   0.5   |   6    |          2          |         24         |     26      |
> | NarrativeQA |   1.5   |   19   |          6          |         76         |     79      |
> |  SQUAD1.1   |  2.25   |   33   |          9          |        132         |     135     |
> |  SQUAD2.0   |  2.25   |   34   |          9          |        136         |     139     |
> |    TATQA    |   0.5   |   8    |          2          |         32         |     34      |
> |     Sum     |    8    |  121   |         32          |        484         |     505     |

---

> ### Author Response · Authors · 2024-11-29
> **Summary of response and look forward to the feedback**
>
> We greatly appreciate the thoughtful critique and suggestions from Reviewer TuG5. Below is a summary of our revisions and clarifications based on the provided feedback.
>
> * Comparison of Neighboring Domain Methods. The work most closely related to ours is text normalization, a common industrial approach used to optimize the input for TTS models, which rewriting queries to fit the input requirements of the TTS model without altering their expression.
> We added a text normalization method as a baseline for comparison, which achieved 98% accuracy in the Google Text Normalization Challenge. We present the experimental results in Tables 2, 3, 14, and 15 of the revised manuscript.
>
> * Scalability. We considered both multi-speaker and single-speaker settings, adding two TTS models: MeloTTS and MMS-TTS-ENG.
> The experimental results under the multi-speaker setting are presented in Tables 2 and 3, while the results for the single-speaker setting are shown in Tables 14 and 15.
> The results validate the excellent scalability of our method and demonstrate its unique advantages, bridging the gap in synthesis accuracy between vocoder-based TTS models and autoregressive TTS models.
>
> * Computational Cost. In Table 7 of the revised manuscript, we provide a more detailed discussion on experimental costs to demonstrate the superiority of our method in terms of computational efficiency.
>
> * Discussion on Thresholds. In Table 4, we present the impact of data quality thresholds on downstream tasks, and in Section 5.1 (Dataset), we provide a more detailed explanation of the composition of the training data.
>
> * Human Evaluation
>
> **Generation Quality Evaluation**
>
> We selected 50 samples from Drop, Quoref, Ropes, and NarrativeQA, respectively, and had human annotators evaluate the quality of the responses. We ensured that the input speech did not contain any quality issues in the text.
>
> During the evaluation process, we asked annotators to assess whether the model-generated results were consistent with the reference answers. The experimental results are shown in the table below. We reported the number of samples that passed the verification.
>
>
>
> | Method         	 | Drop 	 | Quoref 	 | Ropes 	 | NarrativeQA 	 |
> |------------------|--------|----------|---------|---------------|
> | Continue+Ours  	 | 8    	 | 32     	 | 24    	 | 19          	 |
> | Golden+Oringai 	 | 7    	 | 29     	 | 25    	 | 21          	 |
> | Golden+Ours    	 | 20   	 | 41     	 | 29    	 | 31          	 |
>
> The method settings are as follows:
>
> * Continue+Ours: The training objective uses the Continue setting, and the data synthesis method uses our proposed approach.
> * Golden+Original: The training objective uses the Golden setting, and the data synthesis method uses the original setting.
> * Golden+Ours: The training objective uses the Golden setting, and the data synthesis method uses our proposed approach.
>
> These settings correspond to the training methods in rows 2–4 of Table 4.
>
> **Data Quality Evaluation**
>
> We conducted a small-scale manual annotation experiment to demonstrate the improvement in data quality achieved by our method under rigorous human-machine interaction evaluation.
>
> Specifically, we randomly selected 100 samples from each of the four datasets: DROP, Quoref, ROPES, and NarrativeQA.  We ensured that these samples had a SIM score below 50 under the original method, which indicates that there are significant differences in linguistic information between the speech data and the original text.
>
> During the annotation process, we instructed annotators to verify whether the speech content conveyed the same meaning as the original text and provide annotation results.  Any annotations completed in less time than the duration of the audio were discarded and re-annotated to ensure the validity of the annotations.
>
> The following are our experimental results, demonstrating that our method significantly improves the linguistic consistency of synthesized speech.
>
> | Method  	  | DROP 	 | Quoref 	 | ROPES 	 | NarrativeQA 	 |
> |:----------:|:------:|:--------:|:-------:|:-------------:|
> | Original 	 |  0  	  |  0   	   |  0   	  |   0      	    |
> |  TN    	   | 34  	  |  47   	  |  52  	  |   49     	    |
> |  Ours   	  | 74  	  |  79   	  |  69  	  |   71     	    |
>
>
> We hope these revisions and clarifications address your concerns and look forward to any additional feedback or questions.

---

### Meta-Review · Area_Chair_tUym · 2024-12-18

**Metareview:**

This paper presents a novel framework for constructing high-quality speech instruction datasets, focusing on query rewriting to enhance the ability of TTS models (Text-to-Speech models) to process and synthesize text instructions. The key contributions include:
- Query Rewriting with Multi-LLM Knowledge Fusion: The paper introduces a query rewriting framework that leverages multiple LLMs to rewrite text instructions, aligning them with typical TTS model training distributions while preserving semantics. This enhances the linguistic alignment between text and generated speech. The framework employs multi-agent annotation using multiple ASR models and embedding-based similarity metrics to validate and ensure high-quality, well-aligned speech output. A knowledge fusion method is used to combine the strengths of various LLMs, addressing challenging rewriting cases and improving the overall model’s ability to handle diverse queries and contexts.
- Efficacy of proposed method is showed empirically: the paper demonstrates that the proposed framework improves data usability and the quality of synthetic speech. This is achieved while minimizing the need for human annotation, making the approach cost-effective and scalable. Experimental results also show that the approach enhances linguistic information preservation and the overall quality of the generated synthetic speech across multiple datasets. The alignment between synthetic speech and text instructions also helps speech language models (LSLMs) follow spoken instructions more effectively.
- To enhance speaker diversity, the method uses an LLM to generate speaker descriptions that control the speaker identity and style in the generated speech, contributing to more natural and varied speech outputs.


Strength of this paper
- cost efficiency and practicality: The paper introduces a cost efficient (compared to human annotation), practical solution to a significant problem in speech instruction dataset creation at scale. It presents a versatile framework that integrates existing LLMs, ASR, and text embedding methods in a plug-and-play manner, without requiring additional training aside from the knowledge fusion component.
- Empirical evidence to support the effectiveness of proposed method: the authors provided experiment results showing improvements in data quality (e.g., WER, SIM, PASS) and downstream task performance (e.g., on NarrativeQA)


Weakness of this paper

Several reviewers raised few concerns/limitations of this paper. By addressing these limitations, the paper could strengthen its experiment and expand impact.
- Insufficient validation and experimental scope: The paper primarily relies on embedding similarity metrics, which could be complemented with human evaluations to better assess the quality of the results. The main result is limited to a single model and dataset, which is insufficient to generalize the approach to other downstream tasks. There is a lack of clarity on experimental setup and additional datasets that were available but not used. Besides, ablation studies are incomplete, particularly regarding comparisons with simpler alternatives (e.g., using raw TTS outputs without the rewriting process, or a wide range of query rewriting methods from adjacent domains); the study is limited to text normalization in TTS, focusing on input text rewriting rather than advancements in speech language models (SLMs) training or model evaluation.
- Marginal Novel: the method is marginal novel, as many components (e.g., TTS for audio/speech data synthesis, ensembling multiple LLMs and agents) are common practices. This work combines these pieces to carefully generate a dataset with good quality. One possibility to make the contribution more novel and generalizable, is to provide thorough analysis or discussion on the success and failure cases, help to readers better understand the method's limitations and future applicability.
- Scalability and dependency concerns: Scalability and the computational costs of the proposed method are not thoroughly discussed. This work is dependent on specific TTS models and ASR models, which raises questions about the approach’s performance with other models.

**Additional Comments On Reviewer Discussion:**

Although	 some of these weakness have been improved / somewhat addressed during rebuttal session, I do feel the session is too short and I would like to see a more comprehensive modification to systematically working on these suggestions. The final review scores didn't change much and most reviewers still gave score below acceptance. Thus I recommend the authors to re-work on these weakness and re-submitting to future conferences.

---

### Decision · Program_Chairs · 2025-01-22

Reject